# MELR: META-LEARNING VIA MODELING EPISODE-LEVEL RELATIONSHIPS FOR FEW-SHOT LEARNING

**Nanyi Fei**
School of Information, Renmin University of China, Beijing, China
`feinanyi@ruc.edu.cn`

**Zhiwu Lu** *
Gaoling School of Artificial Intelligence, Renmin University of China, Beijing, China
Beijing Key Laboratory of Big Data Management and Analysis Methods, Beijing, China
`luzhiwu@ruc.edu.cn`

**Tao Xiang**
University of Surrey
Guildford, Surrey, UK
`t.xiang@surrey.ac.uk`

**Songfang Huang**
Alibaba DAMO Academy
Hangzhou, China
`songfang.hsf@alibaba-inc.com`

## ABSTRACT

Most recent few-shot learning (FSL) approaches are based on episodic training whereby each episode samples few training instances (shots) per class to imitate the test condition. However, this strict adhering to test condition has a negative side effect, that is, the trained model is susceptible to the poor sampling of few shots. In this work, for the first time, this problem is addressed by exploiting inter-episode relationships. Specifically, a novel meta-learning via modeling episode-level relationships (MELR) framework is proposed. By sampling two episodes containing the same set of classes for meta-training, MELR is designed to ensure that the meta-learned model is robust against the presence of poorly-sampled shots in the meta-test stage. This is achieved through two key components: (1) a Cross-Episode Attention Module (CEAM) to improve the ability of alleviating the effects of poorly-sampled shots, and (2) a Cross-Episode Consistency Regularization (CECR) to enforce that the two classifiers learned from the two episodes are consistent even when there are unrepresentative instances. Extensive experiments for non-transductive standard FSL on two benchmarks show that our MELR achieves 1.0%–5.0% improvements over the baseline (i.e., ProtoNet) used for FSL in our model and outperforms the latest competitors under the same settings.

## 1 INTRODUCTION

Deep convolutional neural networks (CNNs) have achieved tremendous successes in a wide range of computer vision tasks including object recognition (Krizhevsky et al., 2012; Simonyan & Zisserman, 2015; Russakovsky et al., 2015; He et al., 2016a), semantic segmentation (Long et al., 2015; Chen et al., 2018), and object detection (Ren et al., 2015; Redmon et al., 2016). For most visual recognition tasks, at least hundreds of labeled training images are required from each class for training a CNN model. However, collecting a large number of labeled training samples is costly and may even be impossible in real-life application scenarios (Antonie et al., 2001; Yang et al., 2012). To reduce the reliance of deep neural networks on large amount of annotated training data, few-shot learning (FSL) has been studied (Vinyals et al., 2016; Finn et al., 2017; Snell et al., 2017; Sung et al., 2018), which aims to recognize a set of novel classes with only a few labeled samples by knowledge transfer from a set of base classes with abundant samples.

---

*Corresponding author.

Recently, FSL has been dominated by meta-learning based approaches (Finn et al., 2017; Snell et al., 2017; Sung et al., 2018; Lee et al., 2019; Ye et al., 2020), which exploit the ample samples from base classes via episodic training. During meta-training, to imitate an $N$-way $K$-shot novel class recognition task, an $N$-way $K$-shot episode/meta-task is sampled in each iteration from the base classes, consisting of a support set and a query set. By setting up the meta-training episodes exactly the same way as the meta-test ones (i.e., $N$-way $K$-shot in the support set), the objective is to ensure that the meta-learned model can generalize to novel tasks. However, this also leads to an unwanted side-effect, that is, the model will be susceptible to the poor sampling of the few shots.

Outlying training instances are prevalence in vision benchmarks which can be caused by various factors such as occlusions or unusual pose/lighting conditions. When trained with ample samples, modern CNN-based recognition models are typically robust against abnormal instances as long as they are not dominant. However, when as few as one shot per class is used to build a classifier for FSL, the poorly-sampled few shots could be catastrophic, e.g., when the cat class is represented in the support set by a single image of a half-occluded cat viewed from behind, it would be extremely hard to build a classifier to recognize cats in the query set that are mostly full-body visible and frontal. Existing episodic-training based FSL models do not offer any solution to this problem. The main reason is that different episodes are sampled randomly and independently. When the cat class is sampled in two episodes, these models are not aware that they are the same class, and thus cannot enforce the classifiers independently learned to be consistent to each other, regardless whether there exist poorly-sampled shots in one of the two episodes.

In this paper, a novel meta-learning via modeling episode-level relationships (MELR) framework is proposed to address the poor sampling problem of the support set instances in FSL. In contrast to the existing episodic training strategy, MELR conducts meta learning over two episodes deliberately sampled to contain the same set of base classes but different instances. In this way, cross-episode model consistency can be enforced so that the meta-learned model is robust against poorly-sampled shots in the meta-test stage. Concretely, MELR consists of two key components: Cross-Episode Attention Module (CEAM) and Cross-Episode Consistency Regularization (CECR). CEAM is composed of a cross-episode transformer which allows the support set instances to be examined through attention so that unrepresentative support samples can be identified and their negative effects alleviated (especially for computing class prototypes/centers). CECR, on the other hand, exploits the fact that since the two episodes contain the same set of classes, the obtained classifiers (class prototypes) should produce consistent predictions regardless whether there are any poorly-sampled instances in the support set and/or which episode a query instance comes from. This consistency is enforced via cross-episode knowledge distillation.

Our main contributions are three-fold: (1) For the first time, the poor sampling problem of the few shots is formally tackled by modeling the episode-level relationships in meta-learning based FSL. (2) We propose a novel MELR model with two cross-episode components (i.e., CEAM and CECR) to explicitly enforce that the classifiers of the same classes learned from different episodes need to be consistent regardless whether there exist poorly-sampled shots. (3) Extensive experiments for non-transductive standard FSL on two benchmarks show that our MELR achieves significant improvements over the baseline ProtoNet (Snell et al., 2017) and even outperforms the latest competitors under the same settings. We will release the code and models soon.

## 2 RELATED WORK

**Few-Shot Learning.** Few-shot learning (FSL) has become topical recently. Existing methods can be generally divided into four groups: (1) Metric-based methods either learn a suitable embedding space for their chosen/proposed distance metrics (e.g., cosine similarity (Vinyals et al., 2016), Euclidean distance (Snell et al., 2017), and a novel measure SEN (Nguyen et al., 2020)) or directly learn a suitable distance metric (e.g., CNN-based relation module (Sung et al., 2018; Wu et al., 2019), ridge regression (Bertinetto et al., 2019), and graph neural networks (Satorras & Estrach, 2018; Kim et al., 2019; Yang et al., 2020)). Moreover, several approaches (Yoon et al., 2019; Li et al., 2019a; Qiao et al., 2019; Ye et al., 2020; Simon et al., 2020) learn task-specific metrics which are adaptive to each episode instead of learning a shared task-agnostic metric space. (2) Model-based methods (Finn et al., 2017; Nichol et al., 2018; Rusu et al., 2019) learn good model initializations on base classes and then quickly adapt (i.e., finetune) them on novel classes with few shots and a

limited number of gradient update steps. (3) Optimization-based methods (Ravi & Larochelle, 2017; Munkhdalai & Yu, 2017; Li et al., 2017) aim to learn to optimize, that is, to meta-learn optimization algorithms suitable for quick finetuning from base to novel classes. (4) Hallucination-based methods (Hariharan & Girshick, 2017; Wang et al., 2018; Schwartz et al., 2018; Li et al., 2020) learn generators on base classes and then hallucinate new novel class data to augment the few shots. Additionally, there are also other methods that learn to predict network parameters given few novel class samples (Qiao et al., 2018; Gidaris & Komodakis, 2019; Guo & Cheung, 2020). Although the metric-based ProtoNet (Snell et al., 2017) is used as our baseline in this paper, our proposed MELR framework can be easily integrated with other episodic-training based methods.

**Modeling Episode-Level Relationships.** In the FSL area, relatively less effort has been made to *explicitly* model the relationships across episodes. For modeling such episode-level relationships, there are two recent examples: (1) LGM-Net (Li et al., 2019b) proposes an inter-task normalization strategy, which applies batch normalization to all support samples across a batch of episodes in each training iteration. (2) Among a batch of episodes, Meta-Transfer Learning (Sun et al., 2019) records the class with the lowest accuracy in each episode and then re-samples 'hard' meta-tasks from the set of recorded classes. In this work, instead of utilizing the relationships implicitly, we propose to model episode-level relationships (MELR) *explicitly* by focusing on episodes with the same set of classes. Furthermore, our MELR is specifically designed to cope with the poor sampling of the few shots – an objective very different from those in (Li et al., 2019b; Sun et al., 2019).

**Attention Mechanism.** Attention mechanism was first proposed by (Bahdanau et al., 2015) for machine translation and has now achieved great success in natural language processing (Vaswani et al., 2017) and computer vision (Xu et al., 2015). An attention module typically takes a triplet (queries, keys, values) as input and learns interactions between queries and key-value pairs according to certain task objectives. It is referred to as self-attention or cross-attention depending on whether keys and queries are the same. Several recent works (Hou et al., 2019; Guo & Cheung, 2020; Ye et al., 2020) have utilized attention mechanism for meta-learning based FSL. CAN (Hou et al., 2019) employs cross-attention between support and query samples to learn better feature representations. AWGIM (Guo & Cheung, 2020) adopts both self- and cross-attention for generating classification weights. FEAT (Ye et al., 2020) only uses self-attention on the class prototypes of the support set. The biggest difference between these methods and our MELR lies in whether attention is modeled within each episode or across episodes. Only MELR allows modeling cross-episode instance attention explicitly so that the meta-learned model can be insensitive to badly-sampled support set instances. In addition, in our MELR, query set instances are also updated using cross-attention whilst existing models such as FEAT only apply attention to prototypes obtained using support set instances. They thus cannot directly handle instance-level anomalies.

## 3 METHODOLOGY

### 3.1 PROBLEM DEFINITION

Let $\mathcal{D}_b = \{(x_i, y_i) | y_i \in \mathcal{C}_b, i = 1, 2, \cdots, N_b\}$ denote an abundant meta-training set from base classes $\mathcal{C}_b$, where $x_i$ is the $i$-th image, $y_i$ denotes the class label of $x_i$, and $N_b$ is the number of images in $\mathcal{D}_b$. Similarly, let $\mathcal{D}_n = \{(x_i, y_i) | y_i \in \mathcal{C}_n, i = 1, 2, \cdots, N_n\}$ denote a few-shot sample set from a set of novel classes $\mathcal{C}_n$ (e.g., $K$-shot means that each novel class has $K$ labeled images and $N_n = K|\mathcal{C}_n|$), where $\mathcal{C}_b \cap \mathcal{C}_n = \emptyset$. We are also given a test set $\mathcal{T}$ from $\mathcal{C}_n$, where $\mathcal{D}_n \cap \mathcal{T} = \emptyset$. By exploiting $\mathcal{D}_b$ and $\mathcal{D}_n$ for training, the objective of few-shot learning (FSL) is to predict the class labels of test images in $\mathcal{T}$.

### 3.2 META-LEARNING BASED FSL

Most FSL methods are based on meta-learning (Vinyals et al., 2016; Finn et al., 2017; Snell et al., 2017; Sung et al., 2018; Lee et al., 2019; Ye et al., 2020), which adopt episodic training on the base class sample set $\mathcal{D}_b$ and test their models over few-shot classification tasks sampled from the novel classes $\mathcal{C}_n$. Concretely, an $N$-way $K$-shot $Q$-query episode $e = (\mathcal{S}_e, \mathcal{Q}_e)$ is generated as follows: (1) We first randomly sample a subset $\mathcal{C}_e$ from base classes $\mathcal{C}_b$ during meta-training (or from novel classes $\mathcal{C}_n$ during meta-test) and re-index it as $\mathcal{C}_e = \{1, 2, \cdots, N\}$. (2) For each class in $\mathcal{C}_e$, $K$ support and $Q$ query images are then randomly sampled to form the support set $\mathcal{S}_e = \{(x_i, y_i) | y_i \in$

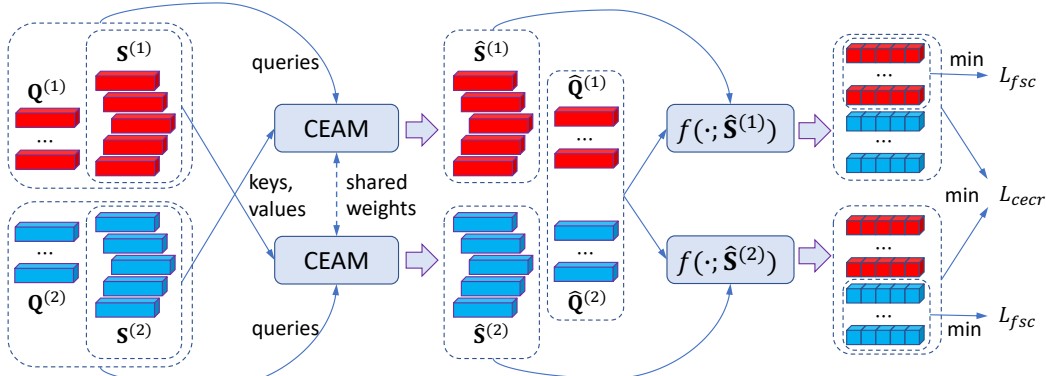

Figure 1: The schematic illustration of the proposed MELR model. It consists of two main components for modeling episode-level relationships: Cross-Episode Attention Module (CEAM) and Cross-Episode Consistency Regularization (CECR). For clarity, only the 5-way 1-shot setting is presented here. Each red/blue cuboid denotes a single instance.

$\mathcal{C}_e, i = 1, 2, \cdots, N \times K\}$ and the query set $\mathcal{Q}_e = \{(x_i, y_i)|y_i \in \mathcal{C}_e, i = 1, 2, \cdots, N \times Q\}$ ($\mathcal{S}_e \cap \mathcal{Q}_e = \emptyset$), respectively.

A meta-learning based FSL approach typically designs a few-shot classification loss over the query set $\mathcal{Q}_e$ for each meta-training episode $e$:

$$L_{fsc}(e) = \mathbb{E}_{(x_i, y_i) \in \mathcal{Q}_e} L(y_i, f(\psi(x_i); \mathcal{S}_e)), \tag{1}$$

where $\psi$ denotes a feature extractor with an output dimension $d$, $f(\cdot\,; \mathcal{S}_e) : \mathbb{R}^d \to \mathbb{R}^N$ can be any scoring function constructed from the support set $\mathcal{S}_e$, and $L(\cdot, \cdot)$ is the classification loss (e.g., widely used cross-entropy). By minimizing the above loss function via back propagation to update the part of the model to be meta-learned (e.g., $\psi$ in ProtoNet), the model is trained over many meta-training episodes and then evaluated on the meta-test episodes.

### 3.3 MODELING EPISODE-LEVEL RELATIONSHIPS (MELR)

In the FSL area, relatively less effort has been made to *explicitly* model the relationships across episodes, and many FSL methods define their loss functions within each episode independently. In contrast, with our Modeling Episode-Level Relationships (MELR), two $N$-way episodes are sampled in each training iteration from exactly the same set of $N$ base classes. Cross-Episode Attention Module (CEAM) and Cross-Episode Consistency Regularization (CECR) are then devised to exploit this type of episode-level relationship explicitly (see Figure 1).

**Cross-Episode Attention Module (CEAM).** We are given two $N$-way $K$-shot $Q$-query episodes $e^{(1)} = (\mathcal{S}_e^{(1)}, \mathcal{Q}_e^{(1)})$ and $e^{(2)} = (\mathcal{S}_e^{(2)}, \mathcal{Q}_e^{(2)})$ sampled from the same subset $\mathcal{C}_e$ of $\mathcal{C}_b$, where $\mathcal{C}_e$ is re-indexed as $\mathcal{C}_e = \{1, 2, \cdots, N\}$, and $e^{(1)} \cap e^{(2)} = \emptyset$. For both episodes, to minimize the negative impact of badly-sampled few shots for a given query instance, we propose CEAM for cross-episode attention modeling, which is detailed below.

Concretely, let $\mathbf{S}^{(1)} = [\psi(x_i)^T; x_i \in \mathcal{S}_e^{(1)}] \in \mathbb{R}^{NK \times d}$ (or $\mathbf{S}^{(2)}$) and $\mathbf{Q}^{(1)} = [\psi(x_i)^T; x_i \in \mathcal{Q}_e^{(1)}] \in \mathbb{R}^{NQ \times d}$ (or $\mathbf{Q}^{(2)}$) denote the feature matrices of support and query samples in $e^{(1)}$ (or $e^{(2)}$), respectively, and let $\mathbf{F}^{(1)} = [\mathbf{S}^{(1)}; \mathbf{Q}^{(1)}] \in \mathbb{R}^{N(K+Q) \times d}$ (or $\mathbf{F}^{(2)} = [\mathbf{S}^{(2)}; \mathbf{Q}^{(2)}]$) be the feature matrix of all samples in $e^{(1)}$ (or $e^{(2)}$). For episode $e^{(1)}$, CEAM takes the triplet $(\mathbf{F}^{(1)}, \mathbf{S}^{(2)}, \mathbf{S}^{(2)})$ as input, which corresponds to the input (queries, keys, values) in a typical attention module:

$$\hat{\mathbf{F}}^{(1)} = \text{CEAM}(\mathbf{F}^{(1)}, \mathbf{S}^{(2)}, \mathbf{S}^{(2)}) = \mathbf{F}^{(1)} + \text{softmax}\left(\frac{\mathbf{F}_Q^{(1)} \mathbf{S}_K^{(2)T}}{\sqrt{d}}\right) \mathbf{S}_V^{(2)}, \tag{2}$$

where the inputs are first linearly mapped into a latent space with the same dimension of the feature space (using projection matrices $\mathbf{W}_Q, \mathbf{W}_K, \mathbf{W}_V \in \mathbb{R}^{d \times d}$):

$$\mathbf{F}_Q^{(1)} = \mathbf{F}^{(1)} \mathbf{W}_Q \in \mathbb{R}^{N(K+Q) \times d}, \tag{3}$$

$$\mathbf{S}_K^{(2)} = \mathbf{S}^{(2)} \mathbf{W}_K \in \mathbb{R}^{NK \times d}, \tag{4}$$

$$\mathbf{S}_V^{(2)} = \mathbf{S}^{(2)} \mathbf{W}_V \in \mathbb{R}^{NK \times d}. \tag{5}$$

Similarly, for episode $e^{(2)}$, we have (analogous to Eq. (2)):

$$\hat{\mathbf{F}}^{(2)} = \text{CEAM}(\mathbf{F}^{(2)}, \mathbf{S}^{(1)}, \mathbf{S}^{(1)}) = \mathbf{F}^{(2)} + \text{softmax}(\frac{\mathbf{F}_Q^{(2)} \mathbf{S}_K^{(1)T}}{\sqrt{d}}) \mathbf{S}_V^{(1)}, \tag{6}$$

where the learnable parameters of fully connected layers (i.e., $\mathbf{W}_Q$, $\mathbf{W}_K$ and $\mathbf{W}_V$) are shared across Eq. (2) and Eq. (6). We can then obtain the transformed support and query embedding matrices in $e^{(1)}$ (or $e^{(2)}$) from $\hat{\mathbf{F}}^{(1)} = [\hat{\mathbf{S}}^{(1)}; \hat{\mathbf{Q}}^{(1)}]$ (or $\hat{\mathbf{F}}^{(2)} = [\hat{\mathbf{S}}^{(2)}; \hat{\mathbf{Q}}^{(2)}]$).

**Cross-Episode Consistency Regularization (CECR).** In our MELR model, CEAM utilizes instance-level attention to alleviate the negative effects of the poor support set instance sampling so that each query set instance can be assigned to the right class with minimal loss. Our CECR is designed to further reduce the model sensitivity to badly-sampled shots in different episodes by forcing the two classifiers learned over the two episodes to produce consistent predictions. There are various options on how to enforce such consistency. CECR adopts a knowledge distillation based strategy as empirically it is the most effective one (see Section 4.3).

Let $f(\cdot; \hat{\mathbf{S}}^{(1)}) : \mathbb{R}^d \to \mathbb{R}^N$ and $f(\cdot; \hat{\mathbf{S}}^{(2)}) : \mathbb{R}^d \to \mathbb{R}^N$ be the scoring functions of the two classifiers constructed from $\hat{\mathbf{S}}^{(1)}$ and $\hat{\mathbf{S}}^{(2)}$, respectively. To determine which classifier/scoring function is stronger, we compute the few-shot classification accuracies of the two classifiers on the merged query samples from both episodes. Concretely, let $\hat{\mathcal{Q}}_e^{(1)} = \{(\hat{\mathbf{q}}_i^{(1)}, y_i^{(1)}) | \hat{\mathbf{q}}_i^{(1)} \in \mathbb{R}^d, i = 1, 2, \cdots, NQ\}$ (or $\hat{\mathcal{Q}}_e^{(2)}$) denote the set of transformed embedding vectors of query samples in $e^{(1)}$ (or $e^{(2)}$), where $\hat{\mathbf{q}}_i^{(1)}$ (or $\hat{\mathbf{q}}_i^{(2)}$) denotes the $i$-th row of the transformed embedding matrix $\hat{\mathbf{Q}}^{(1)}$ (or $\hat{\mathbf{Q}}^{(2)}$). With $\hat{\mathcal{Q}}_e^{(1,2)} = \hat{\mathcal{Q}}_e^{(1)} \cup \hat{\mathcal{Q}}_e^{(2)} = \{(\hat{\mathbf{q}}_i^{(1,2)}, y_i^{(1,2)}), i = 1, 2, \cdots, 2NQ\}$), we are able to compute the few-shot classification accuracies of the two classifiers w.r.t. the ground-truth labels and the corresponding predicted ones (i.e., $\arg\max_j \sigma_j(f(\hat{\mathbf{q}}_i^{(1,2)}; \hat{\mathbf{S}}^{(1)}))$ and $\arg\max_j \sigma_j(f(\hat{\mathbf{q}}_i^{(1,2)}; \hat{\mathbf{S}}^{(2)}))$ ($j = 1, 2, \cdots, N$) for $(\hat{\mathbf{q}}_i^{(1,2)}, y_i^{(1,2)}) \in \hat{\mathcal{Q}}_e^{(1,2)}$, where $\sigma_j(\mathbf{v}) \triangleq \frac{\exp(v_j)}{\sum_{j'=1}^N \exp(v_{j'})}$ ($\mathbf{v} \in \mathbb{R}^N$) is the softmax function).

The classifier with higher accuracy is thus considered to be the stronger one and subsequently used as the teacher classifier for the student to behave consistently with it. Without loss of generality, we assume that $f(\cdot; \hat{\mathbf{S}}^{(1)})$ is stronger than $f(\cdot; \hat{\mathbf{S}}^{(2)})$. We choose the knowledge distillation loss (Hinton et al., 2015) for CECR, which is stated as:

$$L_{cecr}(e^{(1)}, e^{(2)}; T) = \mathbb{E}_{(\hat{\mathbf{q}}_i^{(1,2)}, y_i^{(1,2)}) \in \hat{\mathcal{Q}}_e^{(1,2)}} L'(f(\hat{\mathbf{q}}_i^{(1,2)}; \hat{\mathbf{S}}^{(1)}), f(\hat{\mathbf{q}}_i^{(1,2)}; \hat{\mathbf{S}}^{(2)}); T), \tag{7}$$

where $T$ is the temperature parameter as used in (Hinton et al., 2015). More specifically, when the softmax function $\sigma_j(\mathbf{v}; T) \triangleq \frac{\exp(v_j/T)}{\sum_{j'=1}^N \exp(v_{j'}/T)}$ ($\mathbf{v} \in \mathbb{R}^N$, $j = 1, 2, \cdots, N$) is used, we define $L'(f(\hat{\mathbf{q}}_i^{(1,2)}; \hat{\mathbf{S}}^{(1)}), f(\hat{\mathbf{q}}_i^{(1,2)}; \hat{\mathbf{S}}^{(2)}); T)$ in Eq. (7) with the cross-entropy loss:

$$\begin{aligned}
&L'(f(\hat{\mathbf{q}}_i^{(1,2)}; \hat{\mathbf{S}}^{(1)}), f(\hat{\mathbf{q}}_i^{(1,2)}; \hat{\mathbf{S}}^{(2)}); T) \\
&= -\sum_{j=1}^N \sigma_j(f(\hat{\mathbf{q}}_i^{(1,2)}; \hat{\mathbf{S}}^{(1)}); T) \log\left(\sigma_j(f(\hat{\mathbf{q}}_i^{(1,2)}; \hat{\mathbf{S}}^{(2)}); T)\right).
\end{aligned} \tag{8}$$

Note that we cut off the gradients over $f(\cdot; \hat{\mathbf{S}}^{(1)})$ when back-propagating since the output of the teacher scoring function is treated as the soft target for the student.

---

**Algorithm 1** MELR-based FSL

---

**Input:** Our MELR model with the set of all parameters $\Theta$
       The base class sample set $\mathcal{D}_b$
       Hyper-parameters $\lambda, T$
**Output:** The learned model
 1: **for all** iteration = 1, 2, $\cdots$, MaxIteration **do**
 2:     Randomly sample $e^{(1)}$ and $e^{(2)}$ from $\mathcal{D}_b$, satisfying that $\mathcal{C}_e^{(1)} = \mathcal{C}_e^{(2)}, e^{(1)} \cap e^{(2)} = \emptyset$;
 3:     Compute $\hat{\mathbf{F}}^{(1)}$ for $e^{(1)}$ using CEAM with Eq. (2), and obtain $\hat{\mathbf{F}}^{(2)}$ with Eq. (6) similarly.
 4:     Compute $L_{fsc}(e^{(1)})$ and $L_{fsc}(e^{(2)})$ with Eq. (9), respectively;
 5:     Construct $\hat{\mathcal{Q}}_e^{(1,2)} = \hat{\mathcal{Q}}_e^{(1)} \cup \hat{\mathcal{Q}}_e^{(2)}$ based on the two episodes;
 6:     Determine the teacher episode $e^{(t)}$ and the student $e^{(s)}$ by computing the few-shot classification accuracies of the two classifiers within $e^{(1)}$ and $e^{(2)}$, respectively;
 7:     Compute the CECR loss $L_{cecr}(e^{(t)}, e^{(s)}; T)$ with Eq. (7);
 8:     Compute the total loss $L_{total}$ with Eq. (10);
 9:     Compute the gradients $\nabla_\Theta L_{total}$;
10:     Update $\Theta$ using stochastic gradient descent;
11: **end for**
12: **return** The learned model.

---

### 3.4 MELR-BASED FSL ALGORITHM

As we have mentioned above, in each training iteration, we randomly sample two $N$-way $K$-shot $Q$-query episodes $e^{(1)} = (\mathcal{S}_e^{(1)}, \mathcal{Q}_e^{(1)})$ and $e^{(2)} = (\mathcal{S}_e^{(2)}, \mathcal{Q}_e^{(2)})$, which must have exactly the same set of classes but with different instances. We first transform the feature embeddings with Cross-Episode Attention Module (CEAM) and then compute the few-shot classification loss adopting ProtoNet (Snell et al., 2017) for both episodes ($e \in \{e^{(1)}, e^{(2)}\}$):

$$
\begin{aligned}
L_{fsc}(e) &= \mathbb{E}_{(\hat{\mathbf{q}}_i, y_i) \in \hat{\mathcal{Q}}_e} L(y_i, f_{\text{ProtoNet}}(\hat{\mathbf{q}}_i; \hat{\mathbf{S}})) \\
&= \mathbb{E}_{(\hat{\mathbf{q}}_i, y_i) \in \hat{\mathcal{Q}}_e} - \log \sigma_{y_i}(f_{\text{ProtoNet}}(\hat{\mathbf{q}}_i; \hat{\mathbf{S}})).
\end{aligned}
\tag{9}
$$

Next we determine the stronger/teacher episode to compute the Cross-Episode Consistency Regularization (CECR) loss between the two episodes. The total loss for MELR is finally given by:

$$
L_{total} = \frac{1}{2}(L_{fsc}(e^{(1)}) + L_{fsc}(e^{(2)})) + \lambda L_{cecr}(e^{(t)}, e^{(s)}; T),
\tag{10}
$$

where $e^{(t)} \in \{e^{(1)}, e^{(2)}\}$ denotes the teacher episode, and $e^{(s)} \in \{e^{(1)}, e^{(2)}\}$ is the student.

By combining CEAM and CECR for episodic training, our MELR-based FSL algorithm is summarized in Algorithm 1. Once learned, with the optimal model found by our algorithm, we randomly sample multiple $N$-way $K$-shot meta-test episodes from $\mathcal{C}_n$ for evaluation. In other words, the episode-level relationship is only exploited during meta-training.

## 4 EXPERIMENTS

### 4.1 DATASETS AND SETTINGS

**Datasets.** Two widely-used benchmarks are selected: (1) ***mini*ImageNet** (Vinyals et al., 2016): It contains 100 classes from ILSVRC-12 (Russakovsky et al., 2015). Each class has 600 images. We split it into 64 training classes, 16 validation classes and 20 test classes, as in (Ravi & Larochelle, 2017). (2) ***tiered*ImageNet** (Ren et al., 2018): It is a larger subset of ILSVRC-12, containing 608 classes and 779,165 images in total. We split it into 351 training classes, 97 validation classes and 160 test classes, as in (Ren et al., 2018). All images of the two datasets are resized to $84 \times 84$.

**Evaluation Protocols.** The 5-way 5-shot/1-shot settings are used. Each test episode $e^{(test)} = (\mathcal{S}_e^{(test)}, \mathcal{Q}_e^{(test)})$ has 5 classes randomly sampled from the test split, with 5 or 1 shots and 15 queries per class. We thus have $N = 5$, $K = 5$ or 1, $Q = 15$ as in previous works. Although we meta-train

Table 1: Comparative results of standard FSL on two benchmark datasets. The average 5-way few-shot classification accuracies (%, top-1) along with the 95% confidence intervals are reported.

| Method | Backbone | *mini*ImageNet | | *tiered*ImageNet | |
|---|---|---|---|---|---|
| | | 1-shot | 5-shot | 1-shot | 5-shot |
| MatchingNet (Vinyals et al., 2016) | Conv4-64 | $43.56 \pm 0.84$ | $55.31 \pm 0.73$ | – | – |
| ProtoNet[†] (Snell et al., 2017) | Conv4-64 | $52.78 \pm 0.45$ | $71.26 \pm 0.36$ | $53.82 \pm 0.48$ | $71.77 \pm 0.41$ |
| MAML (Finn et al., 2017) | Conv4-64 | $48.70 \pm 1.84$ | $63.10 \pm 0.92$ | $51.67 \pm 1.81$ | $70.30 \pm 0.08$ |
| RelationNet (Sung et al., 2018) | Conv4-64 | $50.40 \pm 0.80$ | $65.30 \pm 0.70$ | $54.48 \pm 0.93$ | $71.32 \pm 0.78$ |
| IMP (Allen et al., 2019) | Conv4-64 | $49.60 \pm 0.80$ | $68.10 \pm 0.80$ | – | – |
| DN4 (Li et al., 2019c) | Conv4-64 | $51.24 \pm 0.74$ | $71.02 \pm 0.64$ | – | – |
| PARN (Wu et al., 2019) | Conv4-64 | $55.22 \pm 0.84$ | $71.55 \pm 0.66$ | – | – |
| PN+rot (Gidaris et al., 2019) | Conv4-64 | $53.63 \pm 0.43$ | $71.70 \pm 0.36$ | – | – |
| CC+rot (Gidaris et al., 2019) | Conv4-64 | $54.83 \pm 0.43$ | $71.86 \pm 0.33$ | – | – |
| Centroid (Afrasiyabi Arman, 2020) | Conv4-64 | $53.14 \pm 1.06$ | $71.45 \pm 0.72$ | – | – |
| Neg-Cosine (Liu et al., 2020) | Conv4-64 | $52.84 \pm 0.76$ | $70.41 \pm 0.66$ | – | – |
| FEAT (Ye et al., 2020) | Conv4-64 | $55.15 \pm 0.20$ | $71.61 \pm 0.16$ | – | – |
| MELR (ours) | Conv4-64 | $\mathbf{55.35 \pm 0.43}$ | $\mathbf{72.27 \pm 0.35}$ | $\mathbf{56.38 \pm 0.48}$ | $\mathbf{73.22 \pm 0.41}$ |
| ProtoNet[†] (Snell et al., 2017) | Conv4-512 | $53.52 \pm 0.43$ | $73.34 \pm 0.36$ | $55.52 \pm 0.48$ | $74.07 \pm 0.40$ |
| MAML (Finn et al., 2017) | Conv4-512 | $49.33 \pm 0.60$ | $65.17 \pm 0.49$ | $52.84 \pm 0.56$ | $70.91 \pm 0.46$ |
| Relation Net (Sung et al., 2018) | Conv4-512 | $50.86 \pm 0.57$ | $67.32 \pm 0.44$ | $54.69 \pm 0.59$ | $72.71 \pm 0.43$ |
| PN+rot (Gidaris et al., 2019) | Conv4-512 | $56.02 \pm 0.46$ | $74.00 \pm 0.35$ | – | – |
| CC+rot (Gidaris et al., 2019) | Conv4-512 | $56.27 \pm 0.43$ | $74.30 \pm 0.33$ | – | – |
| MELR (ours) | Conv4-512 | $\mathbf{57.54 \pm 0.44}$ | $\mathbf{74.37 \pm 0.34}$ | $\mathbf{60.26 \pm 0.51}$ | $\mathbf{77.25 \pm 0.40}$ |
| ProtoNet[†] (Snell et al., 2017) | ResNet-12 | $62.41 \pm 0.44$ | $80.49 \pm 0.29$ | $69.63 \pm 0.53$ | $84.82 \pm 0.36$ |
| TADAM (Oreshkin et al., 2018) | ResNet-12 | $58.50 \pm 0.30$ | $76.70 \pm 0.38$ | – | – |
| MetaOptNet (Lee et al., 2019) | ResNet-12 | $62.64 \pm 0.61$ | $78.63 \pm 0.46$ | $65.99 \pm 0.72$ | $81.56 \pm 0.63$ |
| MTL (Sun et al., 2019) | ResNet-12 | $61.20 \pm 1.80$ | $75.50 \pm 0.80$ | $65.62 \pm 1.80$ | $80.61 \pm 0.90$ |
| AM3 (Xing et al., 2019) | ResNet-12 | $65.21 \pm 0.49$ | $75.20 \pm 0.36$ | $67.23 \pm 0.34$ | $78.95 \pm 0.22$ |
| Shot-Free (Ravichandran et al., 2019) | ResNet-12 | $59.04 \pm 0.43$ | $77.64 \pm 0.39$ | $66.87 \pm 0.43$ | $82.64 \pm 0.43$ |
| Neg-Cosine (Liu et al., 2020) | ResNet-12 | $63.85 \pm 0.81$ | $81.57 \pm 0.56$ | – | – |
| Distill (Tian et al., 2020) | ResNet-12 | $64.82 \pm 0.60$ | $82.14 \pm 0.43$ | $71.52 \pm 0.69$ | $86.03 \pm 0.49$ |
| DSN-MR (Simon et al., 2020) | ResNet-12 | $64.60 \pm 0.72$ | $79.51 \pm 0.50$ | $67.39 \pm 0.82$ | $82.85 \pm 0.56$ |
| DeepEMD (Zhang et al., 2020) | ResNet-12 | $65.91 \pm 0.82$ | $82.41 \pm 0.56$ | $71.16 \pm 0.87$ | $86.03 \pm 0.58$ |
| FEAT (Ye et al., 2020) | ResNet-12 | $66.78 \pm 0.20$ | $82.05 \pm 0.14$ | $70.80 \pm 0.23$ | $84.79 \pm 0.16$ |
| MELR (ours) | ResNet-12 | $\mathbf{67.40 \pm 0.43}$ | $\mathbf{83.40 \pm 0.28}$ | $\mathbf{72.14 \pm 0.51}$ | $\mathbf{87.01 \pm 0.35}$ |

our MELR with two episodes in each training iteration, we still evaluate it over test episodes one by one, strictly following the standard setting. Moreover, since no cross-episode relationships can be used when testing, we take $(\mathbf{F}^{(test)}, \mathbf{S}^{(test)}, \mathbf{S}^{(test)})$ as the input of CEAM. Note that the meta-test process is *non-transductive* since the embedding of each query sample in $e^{(test)}$ is independently updated using the keys and values coming from the support set. We report average 5-way classification accuracy (%, top-1) over 2,000 test episodes as well as the 95% confidence interval.

**Implementation Details.** Our MELR algorithm adopts Conv4-64 (Vinyals et al., 2016), Conv4-512 and ResNet-12 (He et al., 2016b) as the feature extractors $\psi$ for fair comparison with published results. The output feature dimensions of Conv4-64, Conv4-512 and ResNet-12 are 64, 512, and 640, respectively. To accelerate the entire training process, we pre-train all three backbones on the training split of each dataset as in many previous works (Zhang et al., 2020; Ye et al., 2020; Simon et al., 2020). We use data augmentation during pre-training (as well as meta-training with ResNet-12 on *mini*ImageNet). For ResNet-12, the stochastic gradient descent (SGD) optimizer is employed with the initial learning rate of 1e-4, the weight decay of 5e-4, and the Nesterov momentum of 0.9. For Conv4-64 and Conv4-512, the Adam optimizer (Kingma & Ba, 2015) is adopted with the initial learning rate of 1e-4. The hyper-parameters $\lambda$ and $T$ are respectively selected from $\{0.02, 0.05, 0.1, 0.2\}$ and $\{16, 32, 64, 128\}$ according to the validation performances of our MELR algorithm (see Appendix A.5 for more details). The code and models will be released soon.

## 4.2 MAIN RESULTS

We compare our MELR with the representative/state-of-the-art methods for standard FSL on the two benchmark datasets in Table 1. Note that we re-implement our baseline (i.e., ProtoNet, denoted with [†]) by sampling two episodes in each training iteration since it is still considered as a strong FSL approach especially when the backbone is deep. We can observe from Table 1 that: (1) Meth-

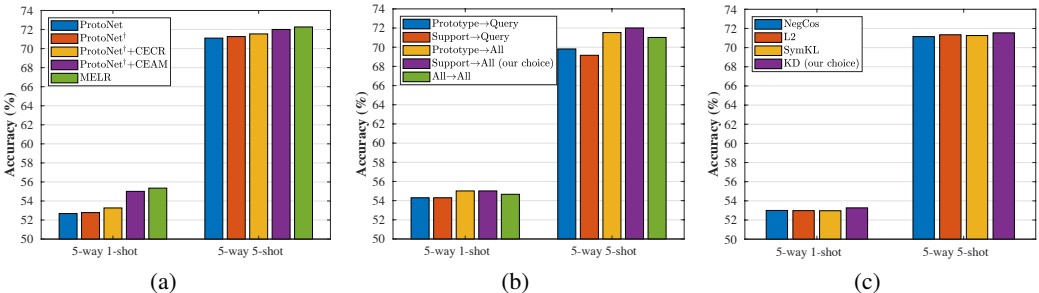

Figure 2: (a) Ablative results for our MELR. (b) Comparison among different ProtoNet$^\dagger$+CEAM alternatives. (c) Comparison among different ProtoNet$^\dagger$+CECR alternatives. All results are obtained under the 5-way 1-shot and 5-shot settings on *mini*ImageNet with Conv4-64 as the feature extractor.

ods trained with ResNet-12 generally perform better than those employing shallower backbones. Also, methods trained with Conv4-512 generally perform better than those employing Conv4-64 even though the two backbones are of the same depth. This is expected because deeper and wider backbones have better representation learning abilities. (2) Our MELR achieves the new state-of-the-art performance on both benchmarks under all settings. Particularly, the improvements over the baseline (i.e., ProtoNet$^\dagger$) range from 1.0% to 5.0%, which clearly validates the effectiveness and the strong generalization ability of our MELR. (3) On both benchmarks, the improvements obtained by our MELR over ProtoNet$^\dagger$ under the 1-shot setting are significantly larger than those under the 5-shot setting. This demonstrates the superior performance of our MELR for FSL with less shots. Again this is expected: FSL with less shots is more likely to suffer from the poor sampling of the few support instances; such a challenging problem is exactly what our MELR is designed for.

### 4.3 FURTHER EVALUATION

**Ablation Study.** To demonstrate the contributions of each cross-episode learning objective in our MELR, we conduct experiments on *mini*ImageNet by adding these learning objectives to the baseline (one at a time) under the 5-way 1-shot and 5-shot settings. Note that ProtoNet without $^\dagger$ is trained with one episode in each training iteration while ProtoNet$^\dagger$ is trained with two episodes per iteration. The ablation study results in Figure 2(a) show that: (1) Increasing mini-batch size helps little for ProtoNet, indicating that our MELR benefits from two cross-episode objectives rather than doubling the mini-batch size. (2) CEAM or CECR alone clearly improves the performance of the baseline model and CEAM appears to be more beneficial to FSL than CECR. (3) The combination of the two cross-episode learning objectives in our full model (i.e., MELR) achieves further improvements, suggesting that these two learning objectives are complementary to each other. Moreover, in Appendix A.2, we conduct more ablative experiments when the attention module is applied within each episode, validating the necessity of cross-episode attention.

**Comparison to CEAM Alternatives.** As we have described, our CEAM takes the support samples as 'keys' and 'values', and all samples in one episode as 'queries' for attention module (denoted as Support→All). We can also input prototypes (mean representations of support samples from the same class) as 'keys' and 'values' or input only query samples as 'queries' for attention. This results in other three alternatives of CEAM: Prototype→Query, Support→Query, and Prototype→All. Note that under the 5-way 1-shot setting, Prototype→Query is equal to Support→Query and Prototype→All is the same as Support→All. Additionally, we compare to All→All: inputting all samples from the other episode as 'keys' and 'values' for CEAM when training but still testing as Support→All (not violating the non-transductive setting). From the comparative results of different choices in Figure 2(b), we can see that Support→All is the best for CEAM. Moreover, All→All works worse than both Prototype→All and Support→All. One possible explanation is that All→All exploits all query set instances during meta-training but only has access to the support set during meta-test to conform to the inductive learning setting. This mis-match reduces its effectiveness.

**Comparison to CECR Alternatives.** Our consistency regularization loss $L_{cecr}$ in Eq. (7) is defined with the knowledge distillation (KD) loss, which can be easily replaced by the negative cosine similarity (NegCos), symmetric Kullback–Leibler divergence (symKL), or the L2 distance (see Ap-

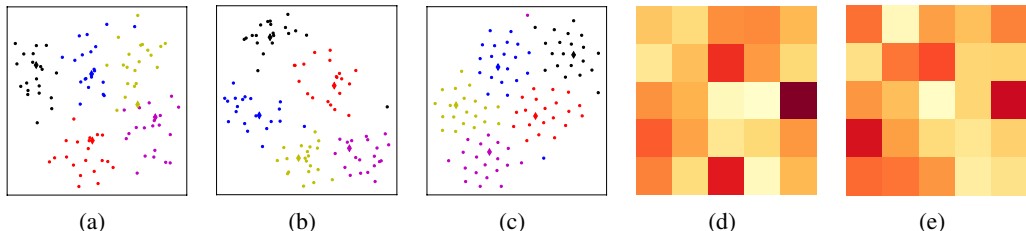

Figure 3: (a) – (c) Visualizations of data distributions obtained by ProtoNet[†], ProtoNet[†]+CEAM, and ProtoNet[†]+CEAM+CECR (i.e., our MELR) for the same meta-test episode, respectively. (d) – (e) Visualizations of attention maps for query and support sets, respectively. All results are obtained under the 5-way 5-shot setting on *mini*ImageNet with Conv4-64 as the feature extractor.

pendix A.1 for more details). The results obtained by ProtoNet[†]+CECR using different consistency losses are shown in Figure 2(c). It can be seen that ProtoNet[†]+KD performs slightly better than other implementations. We thus choose KD as our CECR loss.

**Visualizations of Data Distributions and Attention Maps.** MELR is designed to alleviate the negative effects of poorly-sampled few shots. To validate this, we further provide some visualization results in Figure 3. (1) We sample one episode in the test split of *mini*ImageNet under the 5-way 5-shot setting and obtain the embeddings of all images using the trained models of ProtoNet[†], ProtoNet[†]+CEAM, and our MELR (i.e., ProtoNet[†]+CEAM+CECR), respectively. We then apply t-SNE to project these embeddings into a 2-dimensional space in Figure 3(a) – 3(c). Across three subfigures, samples with the same color belong to the same class and diamonds are class prototypes/centers. We can observe that adding CEAM makes the distribution of different classes more separable (see Figure 3(b) vs. 3(a)), validating the effectiveness of our CEAM. Moreover, the embeddings obtained by our MELR are clearly much more evenly distributed with the prototypes right in the center, indicating less outlying instances (see Figure 3(c) vs. 3(b)). This shows that when CECR is combined with CEAM, those badly-sampled shots in Figure 3(a) are now pulled back to the center of the class distributions. (2) We also visualize the attention maps over the meta-test episode using our trained MELR. Since we take all samples in the episode as 'queries', and support samples as 'keys' and 'values' for CEAM when meta-testing, each of 100 samples has a 25-dimensional weight vector under the 5-way 5-shot setting. For each weight vector, we average the weights of the same class and obtain a 5-dimensional vector. For 75 query samples, we average the vectors of samples with the same class, resulting in a $5 \times 5$ instance attention map (see Figure 3(d)). Similarly, we obtain the attention map for 25 support samples (see Figure 3(e)). It can be seen that the two attention maps are very much alike, indicating that support and query sample embeddings are transformed by our CEAM in a similar way, which thus brings performance improvements for FSL.

## 5 CONCLUSION

We have investigated the challenging problem of how to counter the negative effects of badly-sampled few shots for FSL. For the first time, we propose to exploit the underlying relationships between training episodes with identical sets of classes explicitly for meta-learning. This is achieved by two key components: CEAM is designed for neutralizing unrepresentative support set instances, and CECR is to enforce the prediction consistency of few-shot classifiers obtained in the two episodes. Extensive experiments for non-transductive standard FSL on two benchmarks show that our MELR achieves 1.0%–5.0% improvements over the baseline (i.e., ProtoNet) used for FSL in our model and outperforms the latest competitors under the same settings.

ACKNOWLEDGMENTS

This work was supported in part by National Natural Science Foundation of China (61976220 and 61832017), Beijing Outstanding Young Scientist Program (BJJWZYJH012019100020098), Open Project Program Foundation of Key Laboratory of Opto-Electronics Information Processing, Chinese Academy of Sciences (OEIP-O-202006), and Alibaba Innovative Research (AIR) Program.

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

# A APPENDIX

## A.1 DETAILS ABOUT CECR ALTERNATIVES

For the cross-episode consistency regularization (CECR) loss $L_{cecr}$ in Eq. (7), we compare the knowledge distillation (KD) loss to the negative cosine similarity (NegCos), the L2 distance, and the symmetric Kullback-Leibler (KL) divergence (SymKL) in the main paper. Here we give the details about three CECR alternatives. Concretely, let $\sigma^{(1)}(\hat{\mathbf{q}}_i^{(1,2)})$ (or $\sigma^{(2)}(\hat{\mathbf{q}}_i^{(1,2)})$) denote the normalized vector of $f(\hat{\mathbf{q}}_i^{(1,2)}; \hat{\mathbf{S}}^{(1)})$ (or $f(\hat{\mathbf{q}}_i^{(1,2)}; \hat{\mathbf{S}}^{(2)})$) using softmax, we then have

$$
\begin{aligned}
L_{cecr}^{(NegCos)} =& \mathbb{E}_{(\hat{\mathbf{q}}_i^{(1,2)}, y_i^{(1,2)}) \in \hat{\mathcal{Q}}_e^{(1,2)}} \text{NegCos}(f(\hat{\mathbf{q}}_i^{(1,2)}; \hat{\mathbf{S}}^{(1)}), f(\hat{\mathbf{q}}_i^{(1,2)}; \hat{\mathbf{S}}^{(2)})) \\
=& \mathbb{E}_{(\hat{\mathbf{q}}_i^{(1,2)}, y_i^{(1,2)}) \in \hat{\mathcal{Q}}_e^{(1,2)}} - \frac{< \sigma^{(1)}(\hat{\mathbf{q}}_i^{(1,2)}), \sigma^{(2)}(\hat{\mathbf{q}}_i^{(1,2)}) >}{\|\sigma^{(1)}(\hat{\mathbf{q}}_i^{(1,2)})\|_2 \cdot \|\sigma^{(2)}(\hat{\mathbf{q}}_i^{(1,2)})\|_2},
\end{aligned} \tag{11}
$$

$$
\begin{aligned}
L_{cecr}^{(L2)} =& \mathbb{E}_{(\hat{\mathbf{q}}_i^{(1,2)}, y_i^{(1,2)}) \in \hat{\mathcal{Q}}_e^{(1,2)}} \text{L2}(f(\hat{\mathbf{q}}_i^{(1,2)}; \hat{\mathbf{S}}^{(1)}), f(\hat{\mathbf{q}}_i^{(1,2)}; \hat{\mathbf{S}}^{(2)})) \\
=& \mathbb{E}_{(\hat{\mathbf{q}}_i^{(1,2)}, y_i^{(1,2)}) \in \hat{\mathcal{Q}}_e^{(1,2)}} \|\sigma^{(1)}(\hat{\mathbf{q}}_i^{(1,2)}) - \sigma^{(2)}(\hat{\mathbf{q}}_i^{(1,2)})\|_2,
\end{aligned} \tag{12}
$$

$$
\begin{aligned}
L_{cecr}^{(symKL)} =& \mathbb{E}_{(\hat{\mathbf{q}}_i^{(1,2)}, y_i^{(1,2)}) \in \hat{\mathcal{Q}}_e^{(1,2)}} \text{symKL}(f(\hat{\mathbf{q}}_i^{(1,2)}; \hat{\mathbf{S}}^{(1)}), f(\hat{\mathbf{q}}_i^{(1,2)}; \hat{\mathbf{S}}^{(2)}); T), \\
=& \text{KL}(f(\hat{\mathbf{q}}_i^{(1,2)}; \hat{\mathbf{S}}^{(1)}), f(\hat{\mathbf{q}}_i^{(1,2)}; \hat{\mathbf{S}}^{(2)})/T) \\
& + \text{KL}(f(\hat{\mathbf{q}}_i^{(1,2)}; \hat{\mathbf{S}}^{(2)}), f(\hat{\mathbf{q}}_i^{(1,2)}; \hat{\mathbf{S}}^{(1)})/T),
\end{aligned} \tag{13}
$$

where $< \cdot, \cdot >$ is the inner product of two vectors, $T$ is the temperature parameter, and $\text{KL}(\mathbf{u}, \mathbf{v}) = \sum_{j=1}^{N} \sigma_j(\mathbf{u}) \log \frac{\sigma_j(\mathbf{u})}{\sigma_j(\mathbf{v})}$ ($\mathbf{u}, \mathbf{v} \in \mathbb{R}^N$ are two unnormalized scoring vectors, $\sigma$ denotes the softmax function, and $\sigma_j(\mathbf{u})$ denotes the $j$-th element of $\sigma(\mathbf{u})$).

## A.2 MORE ABLATIVE RESULTS

In Table 2, we show more ablative results when the attention module is applied within each episode independently (named as Intra-Episode Attention Module (IEAM)). Concretely, IEAM is used for ProtoNet[†]+IEAM ([†] means that ProtoNet is trained with two episodes in each training iteration for fair comparison) as follows (i = 1, 2):

$$
\hat{\mathbf{F}}^{(i)} = \text{IEAM}(\mathbf{F}^{(i)}, \mathbf{S}^{(i)}, \mathbf{S}^{(i)}) = \mathbf{F}^{(i)} + \text{softmax}(\frac{\mathbf{F}_Q^{(i)} \mathbf{S}_K^{(i)T}}{\sqrt{d}}) \mathbf{S}_V^{(i)}. \tag{14}
$$

We also add our Cross-Episode Consistency Regularization (CECR) for ProtoNet[†]+IEAM+CECR to see the performance when IEAM instead of our CEAM is adopted.

We can see from Table 2 that adding IEAM to the baseline ProtoNet[†] also improves its performance, but IEAM is not as beneficial as our CEAM (see ProtoNet[†]+IEAM vs. ProtoNet[†]+CEAM). When our CECR is applied on top of ProtoNet[†]+IEAM (i.e., ProtoNet[†]+IEAM+CECR), the improvement is rather minor under the 5-way 1-shot setting and the result even gets worse under the 5-shot setting. However, our MELR can still benefit from CECR (see MELR vs. ProtoNet[†]+CEAM), indicating that CECR is not suitable for IEAM and our CEAM is necessary.

Table 2: Comparative results when IEAM is used on *mini*ImageNet. The average 5-way few-shot classification accuracies (%, top-1) along with the 95% confidence intervals are reported.

| Method | Backbone | *mini*ImageNet | |
| | | 1-shot | 5-shot |
| --- | --- | --- | --- |
| ProtoNet[†] (Snell et al., 2017) | Conv4-64 | $52.61 \pm 0.42$ | $71.33 \pm 0.36$ |
| ProtoNet[†]+IEAM | Conv4-64 | $54.83 \pm 0.43$ | $71.81 \pm 0.35$ |
| ProtoNet[†]+IEAM+CECR | Conv4-64 | $54.97 \pm 0.44$ | $71.72 \pm 0.35$ |
| ProtoNet[†]+CEAM (ours) | Conv4-64 | $55.01 \pm 0.43$ | $72.01 \pm 0.35$ |
| MELR (ours) | Conv4-64 | $\mathbf{55.35 \pm 0.43}$ | $\mathbf{72.27 \pm 0.35}$ |

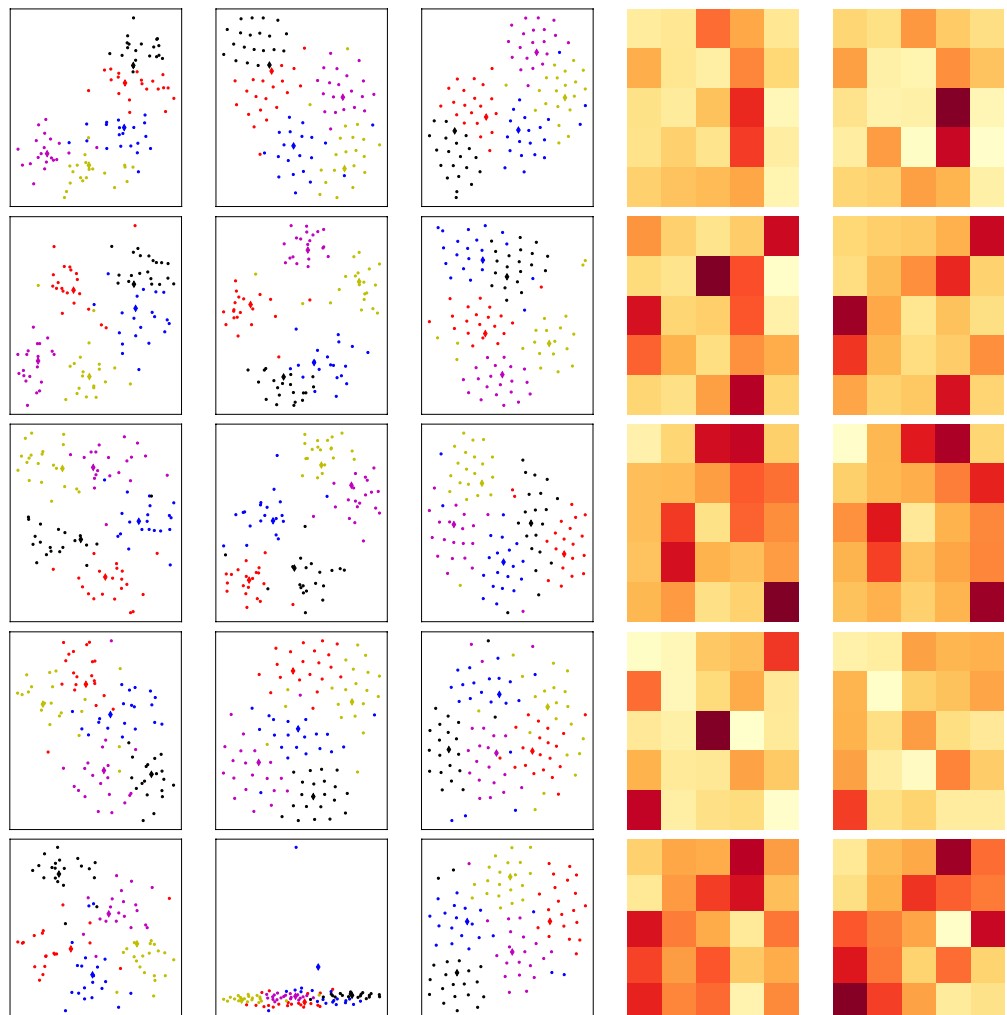

Figure 4: The first three subfigures in each row are the visualizations of data distributions obtained by ProtoNet[†], ProtoNet[†]+CEAM, and ProtoNet[†]+CEAM+CECR (i.e., our MELR) for the same meta-test episode, respectively. The last two subfigures in each row are the visualizations of attention maps for query and support sets, respectively. All results are obtained under the 5-way 5-shot setting on *mini*ImageNet with Conv4-64 as the feature extractor.

## A.3 MORE VISUALIZATION RESULTS

Similar to Section 4.3, we provide more visualization results in Figure 4. (1) We sample five episodes (corresponding to five rows in Figure 4) in the test split of *mini*ImageNet under the 5-way 5-shot setting and visualize the data distributions in the first three columns using the trained models of ProtoNet[†], ProtoNet[†]+CEAM, and our MELR (i.e., ProtoNet[†]+CEAM+CECR), respectively. We can observe that adding CEAM makes the distributions of different classes more separable in the first four rows (see the second column vs. the first column), validating the effectiveness of our CEAM. Moreover, the embeddings obtained by our MELR are clearly much more evenly distributed with the prototypes generally right in the center, indicating less poorly-sampled instances (see the third column vs. the second column). Specifically, ProtoNet[†]+CEAM brings an obvious outlying instance in the last row, but adding CECR stabilizes the training of CEAM. In a word, when CECR is combined with CEAM, the badly-sampled shots can be pulled back to the center of the class distributions. (2) We also visualize the attention maps over each meta-test episode using our trained MELR model. For each of the five episodes, we obtain two $5 \times 5$ attention maps for query and

Table 3: Comparative results of fine-grained FSL on CUB. The average 5-way few-shot classification accuracies (%, top-1) along with the 95% confidence intervals are reported.

| Method | Backbone | CUB | |
| --- | --- | --- | --- |
| | | 1-shot | 5-shot |
| MatchingNet (Vinyals et al., 2016) | Conv4-64 | $61.16 \pm 0.89$ | $72.86 \pm 0.70$ |
| ProtoNet† (Snell et al., 2017) | Conv4-64 | $64.42 \pm 0.48$ | $81.82 \pm 0.35$ |
| MAML (Finn et al., 2017) | Conv4-64 | $55.92 \pm 0.95$ | $72.09 \pm 0.76$ |
| Relation Net (Sung et al., 2018) | Conv4-64 | $62.45 \pm 0.98$ | $76.11 \pm 0.69$ |
| FEAT (Ye et al., 2020) | Conv4-64 | $68.87 \pm 0.22$ | $82.90 \pm 0.15$ |
| ProtoNet†+CEAM (ours) | Conv4-64 | $68.92 \pm 0.50$ | $84.54 \pm 0.32$ |
| MELR (ours) | Conv4-64 | $\mathbf{70.26 \pm 0.50}$ | $\mathbf{85.01 \pm 0.32}$ |

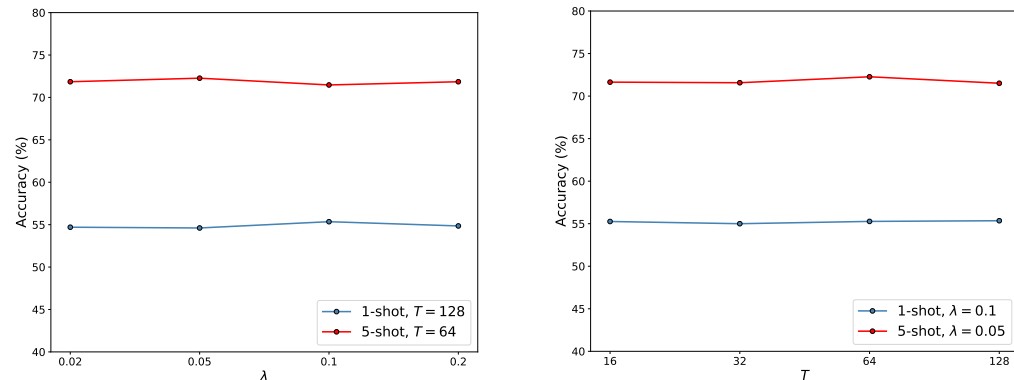

Figure 5: Illustration of hyper-parameter analysis on the *mini*ImageNet dataset. Conv4-64 is used as the feature extractor.

support sets in the last two subfigures of each row, respectively. It can be seen that the two attention maps in each row are very much alike, indicating that support and query sample embeddings are transformed by our CEAM in a similar way, which thus brings performance improvements for FSL.

## A.4 RESULTS FOR FINE-GRAINED FSL

To evaluate our MELR under the fine-grained setting where the poorly-sampled shots may have greater negative impact since the classes are much more close, we conduct experiments on CUB-200-2011 Birds (CUB) (Wah et al., 2011) with Conv4-64 as the feature extractor. CUB has 200 fine-grained classes of birds and 11,788 images in total. We follow (Ye et al., 2020) and split the dataset into 100, 50, and 50 classes for training, validation, and test, respectively. For direct comparison, we also use the pre-trained backbone model released by (Ye et al., 2020), which is pre-trained on the training set. The comparative results in Table 3 show that: (1) Our MELR achieves the best results and improves over the second-best FEAT by 1.4% – 2.1%, validating the effectiveness of MELR under the fine-grained setting. (2) Our ProtoNet†+CEAM alone outperforms all the competitors, and adding CECR into ProtoNet†+CEAM (i.e., our MELR) further brings noticeable improvements (0.5% – 1.3%), indicating that both CEAM and CECR are crucial for fine-grained FSL.

## A.5 ANALYSIS OF HYPER-PARAMETER SENSITIVITY

As we have mentioned in Section 4.1, the hyper-parameters $\lambda$ and $T$ are respectively selected from $\{0.02, 0.05, 0.1, 0.2\}$ and $\{16, 32, 64, 128\}$ according to the validation performance of our MELR algorithm. Concretely, on *mini*ImageNet (with Conv4-64 as the feature extractor), we choose $\lambda = 0.1$ and $T = 128$ under the 5-way 1-shot setting, and choose $\lambda = 0.05$ and $T = 64$ under the 5-way 5-shot setting. In Figure 5, we further present our hyper-parameter analysis on *mini*ImageNet. The results show that our algorithm is quite insensitive to these parameters.

Table 4: Results obtained by varying the number of episodes in each training iteration on *mini*ImageNet (with Conv4-64 as the backbone). The average 5-way classification accuracies (%, top-1) along with the 95% confidence intervals are reported.

| Method | # Episodes | Implement. (1) | | Implement. (2) | |
| --- | --- | --- | --- | --- | --- |
| | | 1-shot | 5-shot | 1-shot | 5-shot |
| MELR | 2 | $\mathbf{55.35 \pm 0.43}$ | $\mathbf{72.27 \pm 0.35}$ | $\mathbf{55.35 \pm 0.43}$ | $\mathbf{72.27 \pm 0.35}$ |
| MELR | 3 | $55.26 \pm 0.44$ | $71.88 \pm 0.35$ | $55.19 \pm 0.43$ | $71.90 \pm 0.35$ |
| MELR | 4 | $55.15 \pm 0.43$ | $71.63 \pm 0.35$ | $55.03 \pm 0.44$ | $71.76 \pm 0.35$ |

Table 5: Comparison regarding the number of parameters ('K' denotes '$\times 10^3$' and 'M' denotes '$\times 10^6$') among various FSL methods.

| Method | Backbone | # Parameters | *mini*ImageNet | |
| --- | --- | --- | --- | --- |
| | | | 1-shot | 5-shot |
| ProtoNet$^{\dagger}$ (Snell et al., 2017) | Conv4-64 | 113.09K | $52.78 \pm 0.45$ | $71.26 \pm 0.36$ |
| PARN (Wu et al., 2019) | Conv4-64 | 405.49K | $55.22 \pm 0.84$ | $71.55 \pm 0.66$ |
| FEAT (Ye et al., 2020) | Conv4-64 | 129.66K | $55.15 \pm 0.20$ | $71.61 \pm 0.16$ |
| MELR (ours) | Conv4-64 | 129.66K | $\mathbf{55.35 \pm 0.43}$ | $\mathbf{72.27 \pm 0.35}$ |
| ProtoNet$^{\dagger}$ (Snell et al., 2017) | ResNet-12 | 12.42M | $62.41 \pm 0.44$ | $80.49 \pm 0.29$ |
| FEAT (Ye et al., 2020) | ResNet-12 | 14.06M | $66.78 \pm 0.20$ | $82.05 \pm 0.14$ |
| MELR (ours) | ResNet-12 | 14.06M | $\mathbf{67.40 \pm 0.43}$ | $\mathbf{83.40 \pm 0.28}$ |

## A.6 Results by Varying the Number of Episodes

We also conduct experiments by varying the number of episodes $N_e$ in each training iteration. For the implementation of CEAM, we have two slightly different choices. Concretely, for each episode $e^{(i)}$ ($i = 1, \cdots, N_e$), the output of CEAM can be defined as:

$$\text{Implement. (1):} \quad \hat{\mathbf{F}}^{(i)} = \frac{1}{N_e - 1} \sum_{j=1,\cdots,N_e, j\neq i} \text{CEAM}(\mathbf{F}^{(i)}, \mathbf{S}^{(j)}, \mathbf{S}^{(j)}); \tag{15}$$

$$\text{Implement. (2):} \quad \hat{\mathbf{F}}^{(i)} = \text{CEAM}(\mathbf{F}^{(i)}, [\mathbf{S}^{(j)}]_{j=1,j\neq i}^{N_e}, [\mathbf{S}^{(j)}]_{j=1,j\neq i}^{N_e}), \tag{16}$$

where $[\mathbf{S}^{(j)}]_{j=1,j\neq i}^{N_e} \in \mathbb{R}^{NK(N_e-1) \times d}$ is the concatenation of $\mathbf{S}^{(j)} \in \mathbb{R}^{NK \times d}$ ($j = 1, \cdots, i-1, i+1, \cdots, N_e$). As for CECR, we determine the episode with the best accuracy as the teacher and distill knowledge to the rest $N_e - 1$ student episodes.

The results on *mini*ImageNet using Conv4-64 in Table 4 show that the performance drops slightly as the number of episodes in each training iteration increases for both implementations. One possible explanation is that too much training data make the model fit better on the training set but fail to improve its generalization ability on novel classes.

## A.7 Comparison Regarding the Number of Parameters

We select several representative/latest FSL models from Table 1 of the main paper and list their numbers of parameters in Table 5. We can observe that: (1) With an extra CEAM in addition to the backbone (Conv4-64 or ResNet-12), our MELR has about $13\% - 15\%$ relatively more parameters than the baseline ProtoNet. Note that CECR (in our MELR) leads to no extra parameters. Considering the (statistically) significant improvements achieved by our MELR over ProtoNet, we think that our MELR is cost-effective because it requires not much additional parameters. (2) The number of MELR's parameters is almost the same as that of FEAT's and is much less than that of PARN's, but our MELR achieves better results than FEAT and PARN, indicating that our MELR is the most cost-effective among these three methods.

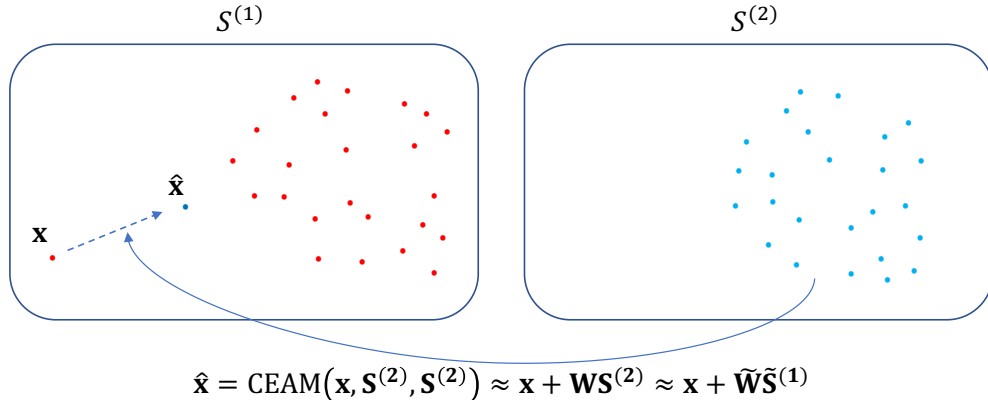

$$\hat{\mathbf{x}} = \text{CEAM}\big(\mathbf{x}, \mathbf{S}^{(2)}, \mathbf{S}^{(2)}\big) \approx \mathbf{x} + \mathbf{W}\mathbf{S}^{(2)} \approx \mathbf{x} + \widetilde{\mathbf{W}}\widetilde{\mathbf{S}}^{(1)}$$

Figure 6: Schematic illustration of our CEAM with a pair of episodes as its input. For easy understanding (but without loss of generality), a toy visual example is considered: only one outlying instance $x$ exists in the support set $\mathcal{S}^{(1)}$ of the first episode, but the support set $\mathcal{S}^{(2)}$ of the second episode is properly sampled. Since $\tilde{\mathcal{S}}^{(1)} = \mathcal{S}^{(1)} \setminus \{x\}$ and $\mathcal{S}^{(2)}$ have similar data distributions (from the same classes), the outlying instance $x$ is pulled back to $\tilde{\mathcal{S}}^{(1)}$ by attending on it with $\mathcal{S}^{(2)}$ (which can not be done by attending on it with $\mathcal{S}^{(1)}$), i.e., its negative effect is mitigated by our CEAM.

Table 6: Results by inputting only support samples as 'queries' into CEAM (denote as 'Support $\to$ Support') on *mini*ImageNet.

| Method | Backbone | 1-shot | 5-shot |
|---|---|---|---|
| ProtoNet[†] | Conv4-64 | $52.61 \pm 0.42$ | $71.33 \pm 0.36$ |
| ProtoNet[†]+CEAM (Support $\to$ Support) | Conv4-64 | $54.79 \pm 0.43$ | $71.40 \pm 0.36$ |
| ProtoNet[†]+CEAM (Support $\to$ All, ours) | Conv4-64 | $\mathbf{55.01 \pm 0.43}$ | $\mathbf{72.01 \pm 0.35}$ |

A.8  SCHEMATIC ILLUSTRATION OF CEAM

To demonstrate how our proposed CEAM can alleviate the negative effect of the poorly sampled shots, we present a schematic illustration of CEAM with a pair of episodes as its input in Figure 6. For easy understanding (but without loss of generality), a toy visual example is considered: only one outlying instance $x$ exists in the support set $\mathcal{S}^{(1)}$ of the first episode, but the support set $\mathcal{S}^{(2)}$ of the second episode is properly sampled. Since the two episodes are sampled from the same set of classes, the data distributions of $\tilde{\mathcal{S}}^{(1)} = \mathcal{S}^{(1)} \setminus \{x\}$ and $\mathcal{S}^{(2)}$ are similar. On one hand, when $\mathcal{S}^{(1)}$ (including the outlier $x$) is used as keys and values to update $\mathcal{S}^{(2)}$, the distribution of $\mathcal{S}^{(2)}$ will not be influenced too much by the outlier $x$ since all the shots in $\mathcal{S}^{(2)}$ are far away from $x$ and the weights on $x$ will be very small. That is, our CEAM is insensitive to few outliers in the keys and values. On the other hand, when $\mathcal{S}^{(1)}$ is transformed based on $\mathcal{S}^{(2)}$, the distribution of $\tilde{\mathcal{S}}^{(1)}$ will be changed little (the data distributions of $\tilde{\mathcal{S}}^{(1)}$ and $\mathcal{S}^{(2)}$ are similar). Particularly, for the outlier $x$, its updated embedding $\hat{\mathbf{x}}$ will be pulled back to $\tilde{\mathcal{S}}^{(1)}$ (i.e., its negative effect is mitigated) since $\hat{\mathbf{x}} = \text{CEAM}(\mathbf{x}, \mathbf{S}^{(2)}, \mathbf{S}^{(2)}) \approx \mathbf{x} + \mathbf{W}\mathbf{S}^{(2)} \approx \mathbf{x} + \tilde{\mathbf{W}}\tilde{\mathbf{S}}^{(1)}$, where $\mathbf{x}$ is the original embedding of $x$, $\mathbf{S}^{(2)}$ and $\tilde{\mathbf{S}}^{(1)}$ are respectively the feature matrices of $\mathcal{S}^{(2)}$ and $\tilde{\mathcal{S}}^{(1)}$, $\mathbf{W}$ and $\tilde{\mathbf{W}}$ are two normalized weight matrices. Note that this cannot be done by attending on $\mathbf{x}$ with $\mathbf{S}^{(1)}$.

Additionally, we provide the results obtained by an extra CEAM alternative in Table 6: only the embeddings of support samples are transformed (denoted as 'Support $\to$ Support') instead of transforming all samples as our choice. We can see from Table 6 and also Figure 2(b) that our choice (i.e., Support $\to$ All) achieves the best results among all CEAM alternatives. One possible explanation for why we resort to updating all samples is that transforming support and query samples into the same embedding space is beneficial to the model learning.

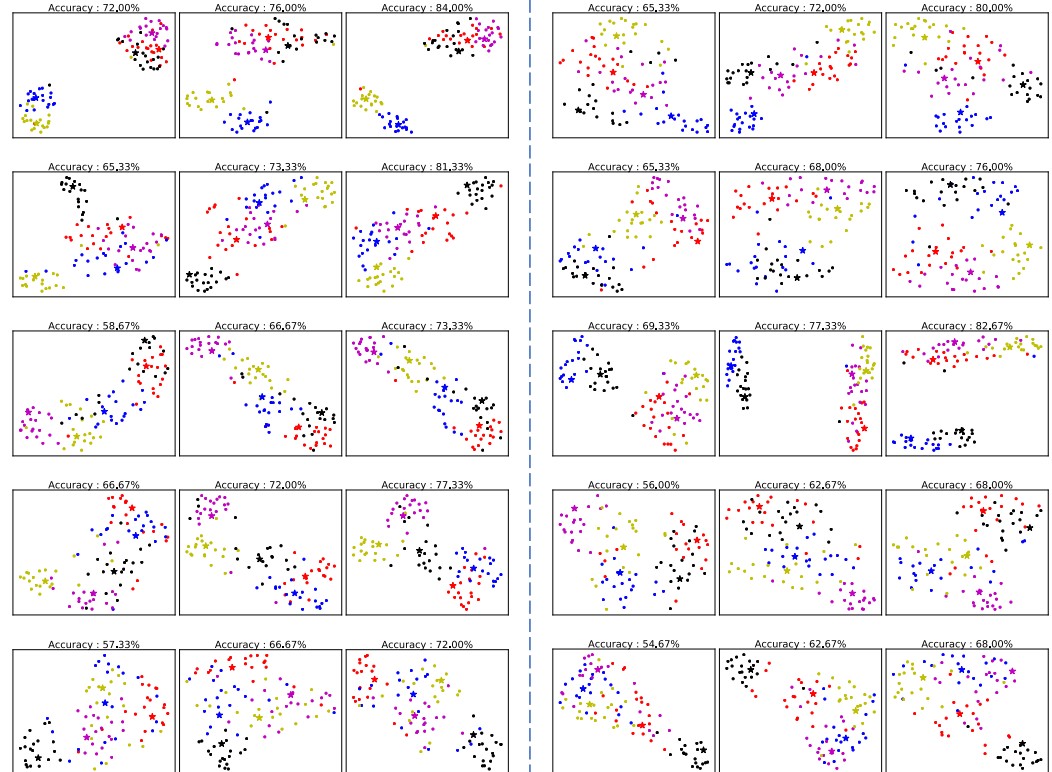

Figure 7: Visualization results of 10 meta-test episodes on *mini*ImageNet under the 5-way 5-shot setting (Conv4-64 is used as the backbone). For each meta-test episode, we visualize three data distributions (from left to right) obtained by ProtoNet[†], ProtoNet[†]+CEAM, and our MELR, respectively. That is, each meta-test episode is denoted by three subfigures and each row has two episodes. In each subfigure, we compute the test accuracy over query samples and present it as the title.

## A.9 SIGNIFICANCE ANALYSIS OF CECR

Since CECR (in our MELR) requires no extra learnable parameters and only computes a loss for consistency constraint, it brings very limited computational cost. Empirically, the meta-training time of MELR is almost the same as that of ProtoNet[†]+CEAM, indicating that the performance improvement over ProtoNet[†]+CEAM is obtained by CECR at an extremely low cost.

To study when CECR has a significant impact on the final FSL performance, we select 10 meta-test episodes from the 2,000 ones used in the evaluation stage and visualize them in Figure 7. Concretely, for each of the 10 selected meta-test episodes, we visualize three data distributions (from left to right) obtained by ProtoNet[†], ProtoNet[†]+CEAM, and our MELR, respectively. That is, each meta-test episode is denoted by a group of three subfigures. In each subfigure, we compute the test accuracy over query samples and present it as the title. We can see that: (1) When the few-shot classification task is hard (i.e., ProtoNet[†] obtains relatively low accuracy), CEAM leads to significant improvements (about $3\% - 9\%$). (2) In the same hard situation, CECR further achieves significant improvements (about $5\% - 8\%$) on top of CEAM and shows its great effect on the final FSL performance. This indicates that CECR and CEAM are complementary to each other in hard situations, and thus both are crucial for solving the poor sampling problem in meta-learning based FSL.

## A.10 RESULTS OF TRANSDUCTIVE FSL

The main difference between standard and transductive FSL is whether query samples are tested one at a time or all simultaneously. As we have mentioned in Section 4.1, we evaluate our MELR model strictly following the *non-transductive* setting for standard FSL since the embedding of each query sample in the test episode is independently transformed using the keys and values coming from the

Table 7: Comparative results of transductive FSL on *mini*ImageNet. The average 5-way few-shot classification accuracies (%, top-1) along with the 95% confidence intervals are reported. We cite the results of the competitors from (Ye et al., 2020).

| Method | Backbone | *mini*ImageNet | |
| | | 1-shot | 5-shot |
| --- | --- | --- | --- |
| **Standard FSL:** | | | |
| ProtoNet[†] (Snell et al., 2017) | Conv4-64 | $52.78 \pm 0.45$ | $71.26 \pm 0.36$ |
| MELR (ours) | Conv4-64 | $55.35 \pm 0.43$ | $72.27 \pm 0.35$ |
| **Transductive FSL:** | | | |
| Semi-ProtoNet (Ren et al., 2018) | Conv4-64 | $55.50 \pm 0.10$ | $71.76 \pm 0.08$ |
| TPN (Liu et al., 2019) | Conv4-64 | $55.51 \pm 0.84$ | $69.86 \pm 0.67$ |
| TEAM (Qiao et al., 2019) | Conv4-64 | $56.57$ | $72.04$ |
| FEAT (Ye et al., 2020) | Conv4-64 | $57.04 \pm 0.16$ | $72.89 \pm 0.20$ |
| ProtoNet[†]+CEAM (ours) | Conv4-64 | $60.30 \pm 0.49$ | $74.28 \pm 0.36$ |
| MELR (ours) | Conv4-64 | $\mathbf{61.67 \pm 0.51}$ | $\mathbf{74.87 \pm 0.35}$ |

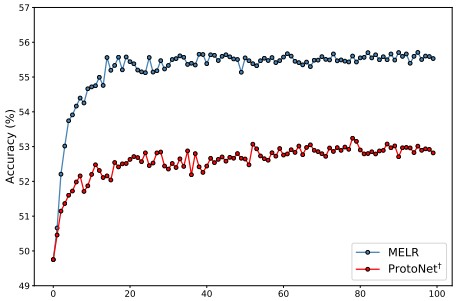 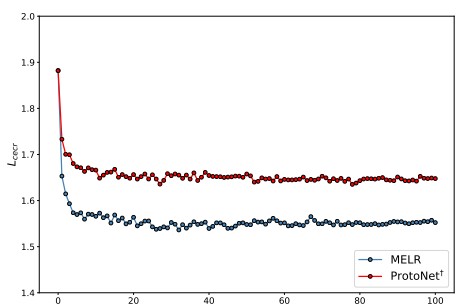

Figure 8: Visualization of the generalization ability of our MELR on the test split of *mini*ImageNet under the 5-way 1-shot setting with Conv4-64 as the backbone. Note that we check the test performance of the model at each *training* epoch (i.e., every 100 training iterations).

support set. However, in this section, we further conduct experiments under the transductive FSL setting to study how well our MELR can make use of the unlabeled query samples.

Concretely, for each meta-test episode $e^{(test)}$, we input all samples (both support and query ones) as keys and values into the trained CEAM:

$$\hat{\mathbf{F}}^{(test)} = \text{CEAM}(\mathbf{F}^{(test)}, \mathbf{F}^{(test)}, \mathbf{F}^{(test)}), \tag{17}$$

such that the relationships of all unlabeled query samples can be taken into consideration. With the transformed embeddings, we make predictions for all query samples based on Semi-ProtoNet (Ren et al., 2018), which utilizes the unlabeled query samples to help construct better class prototypes and then makes predictions similar to ProtoNet. To match the meta-test process, we also make changes to meta-training accordingly. Specifically, for one episode (out of the two) in each training iteration, we use all samples from the other episode as keys and values for CEAM to update all of its own embeddings. This is followed by obtaining the prototypes based on Semi-ProtoNet as well as computing the FSL loss and CECR loss.

The results of transductive FSL on *mini*ImageNet are shown in Table 7. It can be seen that: (1) By utilizing the unlabeled query samples under transductive FSL, our MELR achieves further improvements, as compared to MELR under standard FSL. Particularly, the performance improvement under 1-shot (i.e., 6.3%) is more significant than that under 5-shot (i.e., 2.6%), indicating that exploiting unlabeled query samples brings more benefits to FSL with less labeled support samples. (2) Our MELR achieves the best results among all the transductive FSL methods. Specifically, MELR outperforms FEAT by a large margin (2.0% – 4.6%). Since FEAT also makes predictions based on Semi-ProtoNet, this clearly validates the effectiveness of our MELR under the transductive setting. (3) Our ProtoNet[†]+CEAM alone outperforms all the competitors, and adding CECR into ProtoNet[†]+CEAM (i.e., our MELR is obtained) further brings noticeable improvements (0.6% – 1.4%), indicating that both CEAM and CECR play important roles under transductive FSL.

### A.11 VISUALIZATIONS OF THE GENERALIZATION ABILITY OF MELR

We further provide the visualization of the generalization ability of our MELR during meta-test in Figure 8. Concretely, we randomly sample 1,000 episode pairs from the test split of *mini*ImageNet under the 5-way 1-shot setting, where the two episodes in each pair have identical sets of classes. We then compute the average 5-way classification accuracy over all 2,000 episodes (from the 1,000 episode pairs) and the average $L_{cecr}$ in Eq. (7) over all 1,000 episode pairs at each training epoch. We present the visualization results w.r.t. accuracy and CECR loss in Figure 8. As expected, the accuracy of our MELR is consistently higher than that of our baseline ProtoNet[†]. Moreover, as compared with ProtoNet[†], the CECR loss of our MELR is also lower across the whole training process, indicating that MELR has better performance consistency between two episodes. This provides direct evidence that our CEAM and CECR can boost the generalization ability of the learned model on novel classes.

