# OpenReview forum: "MELR: Meta-Learning via Modeling Episode-Level Relationships for Few-Shot Learning"
_ICLR.cc/2021/Conference — ICLR 2021 Poster_

### Official Review · AnonReviewer2 · 2020-10-24
**Review of MELR: META-LEARNING VIA MODELING EPISODELEVEL RELATIONSHIPS FOR FEW-SHOT LEARNING**

**Rating:** 7
**Confidence:** 4

**Review:**

Summary of Paper:
This paper proposes to improve Prototypical Networks by a method MELR which aims to fix the problem caused by poorly represented classes in sampled episodes and to reap benefits from enforcing cross-episode consistency. The proposed method achieves State of Art results on commonly used benchmarks miniImageNet and tieredImageNet.

Reasons for score:
Overall, I tend towards rejecting this paper. I think cross-episode relationship is an important topic of study in the context of few-shot learning, and the experimental results in this paper are strong. However, I am unconvinced that this ‘hammer’ is the right tool for the ‘nail’ that the authors claim to solve. A more thorough study of the motivating problem and how & why MELR works would greatly improve this paper.

Pros:
1.The proposed method performs well in standard benchmark datasets miniImageNet, tieredImageNet, and CUB200. Assuming correctness of experimental protocol, the improvement over previous methods is significant.
2. Visualization of embedding space by t-SNE lends further credibility to the strong performance. Samples from each class are well clustered yet still disperse.
3. Ablation of hyperparameters and algorithmic alternatives is mostly complete and honestly presented.

Cons:
1. The paper is motivated by the supposed “poorly sampled episodes” problem. While it intuitively makes sense that some data points are more representative than others, whether this has a disparate impact on episode few-shot learning is unclear. In my opinion, the small sample problem in few-shot episodes is no worse than that encountered in standard batch training. In standard supervised learning tasks, batch size as small as 1 has been used successfully to train deep networks given sufficient training epochs. Without empirical or theoretical illustration, I am not convinced that the problem the authors seek to address is a real problem.
2. The reasoning behind equation 2 is unclear. It is not clear why the attention module output in CEAM is added to the embedding F if the goal is to ignore bad examples and emphasize good examples. The choice to ignore class labels when doing attention is also surprising as it doesn’t use the fact that both episodes have the same classes.
3. The proposed method aims to use inter-episode information to stabilize representation learning, hence samples two episodes at a time and apply CEAM and CECR to the joint episode. I don’t see why the authors restrict the method to just two episodes. Classifier consistency should hold transductively across any number of episodes with the same base classes. Thus, you could make the number of episodes an hyperparameter and experimentally verify what is the best choice.
4. Grammar mistakes are common and writing generally lacks polish.

Questions:
	Please address the points in cons above.

Minor points and additional feedback:
Introduction
“even may be impossible” -> may even be impossible
“reliance of deep neural networks on sufficient annotated training data”: you always need “sufficient” data, FSL aims to make fewer data be sufficient.
“Meta-training”, “episode” and “base class” used before definition
“Concretely, MELR consists of two key components: a Cross-Episode Attention Module (CEAM) and a Cross-Episode Consistency Regularization (CECR)” -> remove ‘a’
“two cross-episode components (i.e., CEAM and CECR) to explicitly enforcing” -> to explicitly enforce
Related Work
Very few model-based methods are mentioned but I guess that’s beyond the point here.
“Almost all existing meta-learning based FSL methods ignore the relationships across episodes” -> “Ignore” is probably not true since many works (incl. MAML) frame the problem of Meta-learning as an inter-task learning process (as presented in this review https://arxiv.org/pdf/2004.05439.pdf). There’s also this work (https://arxiv.org/abs/1909.11722) that looks at the role of shots when building episodes during and after meta-training.
Methodology
Too many inline equations. Even with a dedicated definition section there is still a new definition almost every paragraph. Important equations should be made standalone, definitions placed into its own section, and fluff math be removed.
Why does CEAM take S as argument twice? Should they be S_k and S_v instead?
Eqn 2: Is the softmax taken over rows or columns (or both) of F_qS_k^T?

[Post rebuttal] I have increased the score of my review to 7. Below is a copy-pasta of my comments post discussion:

While my original concern about how much sampling affects FSL is still not fully addressed, I think it is not a trivial question to answer and a full exploration of the topic could constitute its own paper. So although I'm not fully convinced about the motivation of this paper, I think the thorough experimental evaluation along with the strong empirical results together warrants publication. From my perspective, a particular important strength of this paper is its ablations. I'm fairly convinced that the presented implementation is likely the best way to implement the idea of cross-episode attention + distillation. I think the baseline proposed by the AC makes sense. It would be great if that could be incorporated into the final version of the paper. A possible explanation for the drop in performance when using 3 or more episodes is due to the relative decrease in episode diversity. Results on wider datasets could corroborate this hypothesis.

---

> ### Author Response · Authors · 2020-11-19
> **Response to AnonReviewer2 – Part 1/2**
>
> Thanks for the reviewer’s constructive comments and suggestions.
>
> **Q1: I am not convinced that the problem the authors seek to address (‘poorly sampled episodes’ problem) is a real problem … In my opinion, the small sample problem in few-shot episodes is no worse than that encountered in standard batch training.** \
> A1: Sorry for the confusion. Indeed, in standard supervised learning, the model updating in each mini-batch iteration suffers little from the poorly sampled mini-batches problem. However, it is very different from the ‘poorly sampled episodes’ problem studied in this paper under the few-shot learning setting. More specifically, in mini-batch based supervised learning, the training instances of the same classes will be sampled in different mini-batches. That is, the overall classification task remains unchanged across different mini-batches. So any mini-batch with poor samples will have a small impact on the final trained model – there are much more iterations/mini-batches that have no outliers to recover the negative effect. In FSL, however, each episode contains a new meta-training task sampled from a pool of seen classes, and in the next episode the task will be completely different. So when there are outliers in each episode, their negative impact on the task must be dealt with immediately. This is because the same model will be deployed for meta-test where *an unseen task is presented only once*, potentially corrupted by outliers. The purpose of our MELR is to meta-learn a mechanism (i.e., meta-learn the ‘poor sampling problem’-solver) so that the negative impact of outliers can be dealt with for any unseen new tasks during meta-test.
>
> **Q2: It is not clear why the attention module output in CEAM is added to the embedding F if the goal is to ignore bad examples and emphasize good examples. The choice to ignore class labels when doing attention is also surprising as it doesn’t use the fact that both episodes have the same classes.** \
> A2: The attention module used in CEAM is essentially a transformer as those used in NLP (e.g., BERT and GPT). So it is designed to update the embeddings of a set of instances through inter-instance attention, such that outliers can be identified and their effect minimized. Adding attention on top of the original embedding of each instance (i.e., adopting a residual structure) is thus a common practice. Moreover, we present the results in the following table when the residual structure is removed from our CEAM (i.e., we remove the original embedding F in Eqs. (2) and (6)). It can be clearly seen that the residual structure is important for solving the original FSL problem (not only the poor sampling problem needs to be solved) since the original embedding contains descriptive information.
>
> |Method&nbsp;|&nbsp;Backbone&nbsp;|&nbsp;5-way 1-shot&nbsp;|&nbsp;5-way 5-shot|
> |-|:-:|:-:|:-:|
> |ProtoNet+CEAM (w/o residual struct.)&nbsp;|Conv4-64|49.45$\pm$0.42|64.81$\pm$0.39|
> |ProtoNet+CEAM (w/ residual struct.)&nbsp;|Conv4-64|55.01$\pm$0.43|72.01$\pm$0.35|
>
> The comment on the class label is an interesting one. We could add a class embedding and combine it to the feature embedding F as input to the attention module. However, this is not needed in our model: the output of our CEAM (i.e., updated embeddings) will be subject to the few-shot classification loss in Eq. (9), which clearly needs to use the class labels of each support set instance. We find that this loss can help to guide the attention module to maintain class separation in the updated embedding space. Moreover, we choose to devise the CECR module that aligns the distributions of predicted scores w.r.t. the two episodes for each query sample. That is, the fact that the two episodes have the same classes is implicitly explored in our CECR.
>
> **Q3: I don’t see why the authors restrict the method to just two episodes.** \
> A3: Thanks for the suggestion. We have added the experiments in Appendix A.6 by varying the number of episodes $N_e$. Concretely, for each episode $e^{(i)}$ ($i = 1, \cdots, N_e$), the output of CEAM is defined as:
> $$\mathbf{\hat{F}}^{(i)} = \frac{1}{N_e-1} \sum_{j = 1, \cdots, N_e, j \ne i} \text{CEAM} (\mathbf{F}^{(i)}, \mathbf{S}^{(j)}, \mathbf{S}^{(j)}).$$
> As for CECR, we determine the episode with the best accuracy and distill knowledge to the rest $N_e-1$ episodes. The results on *mini*ImageNet using Conv4-64 in the following table show that the performance drops slightly as the number of episodes increases. One possible explanation is that too much training data make the model fit better on the training set but fail to improve its generalization ability on novel classes.
>
> |Method&nbsp;|&nbsp;#episodes&nbsp;|&nbsp;5-way 1-shot&nbsp;|&nbsp;5-way 5-shot&nbsp;|
> |-|:-:|:-:|:-:|
> |MELR|2|55.35$\pm$0.43|72.27$\pm$0.35|
> |MELR|3|55.26$\pm$0.44|71.88$\pm$0.35|
> |MELR|4|55.15$\pm$0.43|71.63$\pm$0.35|

---

> > ### Author Response · Authors · 2020-11-19
> > **Response to AnonReviewer2 – Part 2/2**
> >
> > **Q4: Grammar mistakes are common and writing generally lacks polish.** \
> > A4: Thanks for the suggestion. We have carefully polished our paper.
> >
> > **Q5: ‘Ignore’ is probably not true since many works frame the problem of meta-learning as an inter-task learning process.** \
> > A5: Sorry for the confusion. We have changed the claim to ‘In the FSL area, relatively less effort has been made to explicitly model the relationships across episodes.’
> >
> > **Q6: Why does CEAM take $\mathbf{S}$ as argument twice? Should they be $\mathbf{S}_K$ and $\mathbf{S}_V$ instead? Eqn. (2): Is the softmax taken over rows or columns (or both) of $\mathbf{F}_Q \mathbf{S}_K^T$?** \
> > A6: (a) CEAM indeed takes $\mathbf{S}$ as argument twice since one is multiplied with $\mathbf{W}_K$ (resulting in $\mathbf{S}_K$) and the other is multiplied with $\mathbf{W}_V$ (resulting in $\mathbf{S}_V$), which means that the linear projections are done inside CEAM.
> >
> > (b) The softmax is taken over columns in Eqn. (2) since each row is an independent weight vector from other rows.

---

> > > ### Comment · AnonReviewer2 · 2020-11-21
> > > **Thank you for the reply, and some further questions**
> > >
> > > Regarding Q2, I might have been not entirely clear in the original comment. What I want to ask is why was $S^{(2)}$ used as "value" and $F^{(1)}$ as "query" (in the "query key value" language) when computing $F^{(1)}$? If the goal is to reduce outlier effect by selecting the best supports, wouldn't it make more sense to attend on $S^{(1)}$ with $S^{(2)}$ as query?  I see that reviewer 3 also raise a similar question and I think adding a discussion on exactly how "CEAM utilizes instance-level attention to alleviate the negative effects of the poor support set instance sampling" in the paper would help me understand this. Maybe even a toy example to illustrate the concept explicitly?
> > >
> > > Regarding Q3, I'm happy to see the additional results, again I'm a bit surprised by the implementation. For using CEAM with multiple episodes, wouldn't it be more intuitive to concatenate all supports in the "other" episodes and perform attention on that? As in:
> > > \begin{equation}
> > > \hat{\text{F}}^{(i)} = \text{CEAM}(\text{F}^{(i)}, [S^{(j)}]_{j=1,i \neq j}^{N_e}, [S^{(j)}]_{j=1,i \neq j}^{N_e}).
> > > \end{equation}.
> > >
> > > Another point is that the improvement of CECR is within variance of just PN+CEAM in your ablations. Computationally, how expensive is using CECR? Does the performance benefits justify the added complexity. Also, are there usecases where CECR do have a more significant impact on performance (e.g. are there episode settings where MELR significantly outperforms PN+CEAM)?
> > >
> > > If I understand correctly, your reported experimental results are under the non-transductive setting? If so, I think this could be made more clear in the introduction of the paper. I am also curious about whether MELR can improve performance in the transductive setting too.
> > >
> > > I think this paper has potential to be impactful, and the discussions are moving this paper in a good direction.

---

> > > > ### Comment · AnonReviewer1 · 2020-11-22
> > > > **Agree with R2**
> > > >
> > > > I think Reviewer 2 raised an excellent question and I agree with his point of view.
> > > > I'm very curious to see the authors' response to this.

---

> > > > > ### Author Response · Authors · 2020-11-24
> > > > > **RE: Agree with R2**
> > > > >
> > > > > Thanks. Please refer to our response to further comments of AnonReviewer2.

---

> > > > ### Author Response · Authors · 2020-11-24
> > > > **Response to Further Comments of AnonReviewer2**
> > > >
> > > > Thanks for the reviewer’s constructive comments and suggestions.
> > > >
> > > > **1. If the goal is to reduce outlier effect by selecting the best supports, wouldn't it make more sense to attend on $\mathbf{S}^{(1)}$ with $\mathbf{S}^{(2)}$? I think adding a discussion on ... would help me understand this. Maybe even a toy example to illustrate the concept explicitly?** \
> > > > A: To demonstrate how our proposed CEAM can alleviate the negative effect of the poorly sampled shots, we present a schematic illustration of CEAM with a toy visual example (for easy understanding but without loss of generality) in Figure 7 of Appendix A.8. Please refer to the revision for more details.
> > > >
> > > > Additionally, we provide the results obtained by an extra CEAM alternative (as the reviewer suggested) in Table 6 of Appendix A.8. We can see from Table 6 and also Figure 2(b) of the main paper that our choice achieves the best results among all CEAM alternatives. One possible explanation for why we resort to updating all samples is that transforming support and query samples into the same embedding space is beneficial to the model learning.
> > > >
> > > > **2. Regarding Q3, for using CEAM with multiple episodes, wouldn't it be more intuitive to concatenate all supports in the “other” episodes and perform attention on that?** \
> > > > A: Thanks for the suggestion. We name our former implementation as ‘Implement. (1)’ and name the suggested one as ‘Implement. (2)’. The results on *mini*ImageNet using Conv4-64 with ‘Implement. (2)’ are shown below.
> > > >
> > > > | Method | #episodes | 5-way 1-shot | 5-way 5-shot |
> > > > | - | :-: | :-: | :-: |
> > > > | MELR | 2 | 55.35$\pm$0.43 | 72.27$\pm$0.35 |
> > > > | MELR | 3 | 55.19$\pm$0.43 | 71.90$\pm$0.35 |
> > > > | MELR | 4 | 55.03$\pm$0.44 | 71.76$\pm$0.35 |
> > > >
> > > > We have also added these results in Table 4 of Appendix A.6. Please refer to the revision for more details.
> > > >
> > > > **3. Computationally, how expensive is using CECR? Does the performance benefits justify the added complexity. Also, are there usecases where CECR do have a more significant impact on performance (e.g. are there episode settings where MELR significantly outperforms PN+CEAM)?** \
> > > > A: Thanks for the comment. Since CECR (in our MELR) requires no extra learnable parameters and only computes a loss for consistency constraint, it brings very limited computational cost. Empirically, the meta-training time of MELR is almost the same as that of ProtoNet+CEAM, indicating that the improvement over ProtoNet+CEAM is obtained by CECR at an extremely low cost.
> > > >
> > > > To study when CECR has a significant impact on the final FSL performance, we select 10 meta-test episodes from the 2,000 ones used in the evaluation stage and visualize them in Figure 8 (please refer to Appendix A.9 for more details). We can see that: (1) When the few-shot classification task is hard (i.e., ProtoNet obtains relatively low accuracy), CEAM leads to significant improvements (about 3% - 9%). (2) In the same hard situation, CECR further achieves significant improvements (about 5% - 8%) on top of CEAM and shows its great effect on the final FSL performance. This indicates that CECR and CEAM are complementary to each other in hard situations, and thus both are crucial for solving the poor sampling problem in meta-learning based FSL.
> > > >
> > > > **4. If I understand correctly, your reported experimental results are under the non-transductive setting? If so, I think this could be made more clear in the introduction of the paper. I am also curious about whether MELR can improve performance in the transductive setting too.** \
> > > > A: Good suggestion! We did report the experimental results strictly under the non-transductive setting. And we have made it clearer in the abstraction, introduction, and conclusion of the main paper. Moreover, we present comparative results under the transductive FSL setting on *mini*ImageNet in the table below. We can see that our MELR achieves the best results among all the transductive FSL methods. Please also refer to Appendix A.10 for more details.
> > > >
> > > > | Method | Backbone | 5-way 1-shot | 5-way 5-shot |
> > > > | - | :-: | :-: | :-: |
> > > > | Semi-ProtoNet (Ren et al., 2018) | Conv4-64 | 55.50$\pm$0.10 | 71.76$\pm$0.08 |
> > > > | TPN (Liu et al., 2019) | Conv4-64 | 55.51$\pm$0.84 | 69.86$\pm$0.67 |
> > > > | TEAM (Qiao et al., 2019) | Conv4-64 | 56.57 | 72.04 |
> > > > | FEAT (Ye et al., 2020) | Conv4-64 | 57.04$\pm$0.16 | 72.89$\pm$0.20 |
> > > > | ProtoNet+CEAM (ours) | Conv4-64 | 60.30$\pm$0.49 | 74.28$\pm$0.36 |
> > > > | MELR (ours) | Conv4-64 | 61.67$\pm$0.51 | 74.87$\pm$0.35 |

---

### Official Review · AnonReviewer3 · 2020-10-25
**Meta-learning method to alleviate the negative impact of poorly-sampled support sets is proposed. Explanation of why the observed improvements could be achieved will improve the paper's quality.**

**Rating:** 6
**Confidence:** 3

**Review:**

Summary:
- This paper proposes a meta-learning method (MELR) to alleviate the negative impact of poor sampling of support sets.
MLER consists of two main modules. The first module (CEAM) applies the attention mechanism to two sampled episodes and it alleviates the negative impact of badly-sampled instances.
The second module (CECR) enhances the consistency of classifiers obtained by using two episodes to deal with the sensitivity for the badly-sampled instances.
Specifically, to realize this, CECR utilizes a knowledge distillation framework. Experiments with two real-world datasets demonstrate that MELR works well.

Pros:
- This paper proposes a new meta-learning method to alleviate the negative effect of poor sampling of support sets.
- Experimental results show that MELR can improve the baseline method (Propnet).  These results show some evidence of the effectiveness of two proposed modules (CEAM and CECR).

Cons:
- It is not clear why CEAM can alleviate the negative effect of badly-sampled support instances.
- Hyper-parameter candidates used for MELR are not described.

Detailed comments and questions:
- Although empirical results seem to support the effectiveness of CEAM, I do not understand why CEAM alleviates the negative effect of badly-sampled few shots for a given query instance. Can the authors qualitatively explain the reason for this?
- In the last row of fig 4, it seems that CEAM creates obvious outlier instances. Why does this happen?
- How much does the proposed method depend on hyperparameters such as $T$ and $\lambda$?

Minor comments:
- In eqs. (1) and (9), although argmax is taken in the loss function $L$, it is not correct when using the cross-entropy loss.

---

> ### Author Response · Authors · 2020-11-19
> **Response to AnonReviewer3**
>
> Thanks for the reviewer’s constructive comments and suggestions. Our responses are as follows.
>
> **Q1: It is not clear why CEAM can alleviate the negative effect of badly-sampled support instances. Can the authors qualitatively explain the reason for this? In the last row of fig 4 (*now Figure 5 in the revision*), it seems that CEAM creates obvious outlier instances. Why does this happen?** \
> A1: Under the few-shot setting, the biggest negative effect of having an outlying support instance is that the class mean/prototype will be heavily biased by the outlier. This would lead to prototypes of different classes to be close to each other, causing problems for classifying query samples using these prototypes.
>
> Our CEAM is essentially a self-attention based transformer. It transforms the latent embedding of each support set sample by allowing it to examine (attend to) other samples, both from the same class and different classes, in the support set, and update its embedding as a weighted sum of all samples. The weight is determined mostly by similarity or proximity in the original embedding space. Based on this understanding, it is now easy to understand the usefulness as well as the limitation of CEAM for countering the negative effect of bad samples. In particular, the learned transformer is subject to the cross-entropy loss computed on the query using the updated prototypes. This encourages the transformer in CEAM to update the embeddings so that different classes become well separable, therefore largely neutralizing the main negative effect of the outlying samples, as supported by Figure 3(b) vs. 3(a) and more visualizations in Figure 5 in the revision.
>
> However, the effect of CEAM on the relationship of outliers and inliers of the same class is limited, which is determined by the nature of the transformer: the inliers are far away from the outliers so would have limited effect in pulling them toward the inlier majority. This is shown clearly in Figure 5 bottom row – when the outliers are extremely different from the inliers, not being able to pull them close to the inliers would hinder CEAM’s main objective of preventing class overlapping. That is why in this extreme case we need CECR: the predictions based on two sets of prototypes need to be consistent using CECR, which means that the learned embeddings must pull the outliers closer to the inliers. Therefore, both CEAM and CECR are necessary for our full model.
>
> **Q2: Hyper-parameter candidates used for MELR are not described. How much does the proposed method depend on hyper-parameters such as $T$ and $\lambda$?** \
> A2: Thanks. We have added the hyper-parameter candidates in Section 4.1 of the main paper. Concretely, we select $T$ from {16, 32, 64, 128} and $\lambda$ from {0.02, 0.05, 0.1, 0.2} based on the validation performances. Moreover, we have also added a hyper-parameter analysis in Appendix A.5. Our method is shown to be insensitive to the hyper-parameters.
>
> **Q3: In eqs. (1) and (9), although argmax is taken in the loss function, it is not correct when using the cross-entropy loss.** \
> A3: Thanks. We have made the correction in the revision.

---

> > ### Author Response · Authors · 2020-11-24
> > **Additional Response to Q1 of AnonReviewer3**
> >
> > Regarding Q1 ('It is not clear why CEAM can alleviate the negative effect of badly-sampled support instances'), we have added a discussion in Appendix A.8. Concretely, we present a schematic illustration of CEAM with a toy visual example (for easy understanding but without loss of generality) in Figure 7 of Appendix A.8. Please refer to the revision for more details.
> >
> > Thank you again for your constructive review!

---

### Official Review · AnonReviewer1 · 2020-10-25
**Good paper, proper experimental evaluation.**

**Rating:** 6
**Confidence:** 5

**Review:**

### Summary
This paper proposes a way to exploit relationships across tasks in episodic training with the goal of improving the trained models who might be susceptible to poor sampling in for few-shot learning scenarios. The proposed model consists of two components: a cross-attention transformer (CEAM) which is used to observe details across two episodes, and a regularization term (CECR) which imposes that two different instances of the same task (which have the exact same classes) are consistent in terms of prediction. Cross-attention is computed via a scaled-attention transformer using both support and query set. The consistency loss is a knowledge distillation that imposes an agreement on the two episodes. The soft target is chosen among the two predictions selecting the classifier with the highest accuracy.
### Considerations:
- I like the idea of exploiting the information across tasks to improve the performance of episodic meta training. This is an interesting direction that should might definitely help disambiguate in the case of poor sampling.
- The ablation study is accurately performed giving the impression of a careful examination of the components of the model proposed.
- I'm not sure the authors can claim sota results: here some of the latest models that perform best on mini-imagenet https://paperswithcode.com/sota/few-shot-image-classification-on-mini-1 I would prefer to restate the contribution as an improvement of x% over the baseline. It is obvious that sota performance requires higher capacity models such as dense-net. I think that other experiments are needed in order to make the claim of achieving sota, otherwise, if the claim is changed, I'm satisfied with the experiments.
- I suggest the authors moving algorithm 1 in the main paper, maybe replacing the verbose description of each step with actual formulas and pseudocode.
- I think there is still room for improvement on the manuscript.  The paper might be a good contribution to the scientific community, but I'll wait for the authors' response on my doubts before my final decision.

### Questions:
- Q1: Why only considering tasks with the same classes for consistency? Why not considering also partial overlapping of classes across tasks? I guess it is only for simplicity, but it might be beneficial to consider other types of relationships.
- Q2: It is not exactly clear how the meta-test evaluation is performed. I understand that during training you always consider a pair of episodes that are used to transform the features, but how does this translate at meta-test time? Do you always need a pair of episodes? My guess is no, but not using the transformer should change the distribution of the features at test-time and I don't find it trivial to see how this is taken into account. Maybe I'm just misreading the paper. I suggest the authors clarifying this point in the paper.

---

> ### Author Response · Authors · 2020-11-19
> **Response to AnonReviewer1**
>
> We’d like to thank the reviewer for the constructive comments and suggestions. We have accordingly made changes in the revision.
>
> **Q1: I'm not sure the authors can claim SOTA results.** \
> A1: Thanks for pointing this out. To avoid misunderstanding on the SOTA claim, we have now modified the claim in the revision as suggested by the reviewer. Indeed, higher results have been reported elsewhere but achieved with larger backbones, external data, and/or the transductive setting. Under the most standard FSL setting (with three commonly used backbones, no external data, non-transductive), our MELR indeed achieves the best performance on the two benchmarks.
>
> **Q2: I suggest the authors moving algorithm 1 in the main paper.** \
> A2: Thanks for the suggestion. We have moved Algorithm 1 to the main paper and modified the descriptions.
>
> **Q3: Why only considering tasks with the same classes for consistency? Why not considering also partial overlapping of classes across tasks?** \
> A3: Great suggestion! After the ICLR paper submission, we have actually considered episodes with partially overlapped sets of classes. Our main idea is to modify the CECR part but with CEAM unchanged. Concretely, given a pair of randomly sampled episodes, for support samples from the disjointed classes (if any) that come from only one episode, we apply data augmentation (e.g., random crops/horizontal flip) to them so that each class from the two episodes now has two sets of $K$ shots for us to implement CECR. The obtained results under the 5-way 1-shot and 5-shot settings on *mini*ImageNet (with Conv4-64 as the backbone) are 54.72%$\pm$0.43% and 71.51%$\pm$0.35%, respectively. These are good results. However, it seems that exploiting partial overlapping of classes across tasks does not lead to performance improvements over our main results in Table 1 (55.35%$\pm$0.43% for 1-shot and 72.27%$\pm$0.35% for 5-shot). Perhaps new algorithms need to be designed to model other types of cross-episode relationships more effectively – we will leave it to the future work.
>
> **Q4: It is not exactly clear how the meta-test evaluation is performed.** \
> A4: Sorry for the confusion about evaluation protocols. Indeed, only one episode is needed during meta-test, which is the same as previous works. We have stated in Section 4.1 that our MELR is evaluated over meta-test episodes independently (i.e., one episode at a time). Concretely, we apply the trained CEAM within each episode $e^{(test)}$ by inputting the triplet $(\mathbf{F}^{(test)}, \mathbf{S}^{(test)}, \mathbf{S}^{(test)})$, where $\mathbf{S}^{(test)} \in \mathbb{R}^{NK \times d}$ and $\mathbf{F}^{(test)} \in \mathbb{R}^{N(K+Q) \times d}$ are the feature matrices of support samples and all samples in $e^{(test)}$, respectively. That is, for each meta-test episode, we take its own support samples (instead of those from another episode) as keys and values for the trained CEAM. Note that we strictly follow the *non-transductive* evaluation setting since the embedding of each query sample in $e^{(test)}$ is transformed by the trained CEAM independently according to the support sets.

---

> > ### Comment · AnonReviewer1 · 2020-11-21
> > **Question regarding the number of parameters**
> >
> > Thank you for your answers.
> > - I would have expected that using also partially overlapped sets of classes and more episodes (as suggested by R2) would have helped, but I agree that it could be left for future work.
> >
> > - I have a further question, the relative improvement in all the settings wrt to the baseline is noticeable, but the authors should provide the additional number of parameters that the proposed method requires. I think that in order to have a much fairer comparison the baseline should match the number of parameters of MELR.

---

> > > ### Author Response · Authors · 2020-11-22
> > > **Response to AnonReviewer1**
> > >
> > > **Q1: Question regarding the number of parameters.** \
> > > A1: We select several representative/latest FSL models from Table 1 of the main paper and list their numbers of parameters in the table below. The accuracies in this table are obtained on *mini*ImageNet. We can observe that: (1) With an extra CEAM in addition to the backbone (Conv4-64 or ResNet-12), our MELR has about 13-15% relatively more parameters than the baseline ProtoNet. Note that CECR (in our MELR) leads to no extra parameters. Considering the (statistically) significant improvements achieved by our MELR over ProtoNet, we think that our MELR is cost-effective because it requires not much additional parameters. (2) The number of MELR's parameters is almost the same as that of FEAT's and is much less than that of PARN's, but our MELR achieves better results than FEAT and PARN, indicating that our MELR is the most cost-effective among these three methods. Note that FEAT also takes ProtoNet as the baseline. We have added this discussion in Appendix A.7.
> > >
> > > | Method | Backbone | # Parameters | 5-way 1-shot | 5-way 5-shot |
> > > | - | :-: | :-: | :-: | :-: |
> > > | ProtoNet (Snell et al., 2017) | Conv4-64 | 113.09K | $52.78\pm0.45$ | $71.26\pm0.36$ |
> > > | PARN (Wu et al., 2019) | Conv4-64 | 405.49K | $55.22\pm0.84$ | $71.55\pm0.66$ |
> > > | FEAT (Ye et al., 2020) | Conv4-64 | 129.66K | $55.15 \pm 0.20$ | $71.61 \pm 0.16$ |
> > > | MELR (ours) | Conv4-64 | 129.66K | $\bf55.35\pm0.43$ | $\bf72.27\pm0.35$ |
> > > | ProtoNet (Snell et al., 2017) | ResNet-12 | 12.42M | $62.41\pm0.44$ | $80.49\pm0.29$ |
> > > | FEAT (Ye et al., 2020) | ResNet-12 | 14.06M | $66.78\pm0.20$ | $82.05\pm0.14$ |
> > > | MELR (ours) | ResNet-12 | 14.06M | $\bf67.40\pm0.43$ | $\bf83.40\pm0.28$ |

---

> > > > ### Comment · AnonReviewer1 · 2020-11-24
> > > > **Thanks for your response. Further question on ablation study**
> > > >
> > > > Thank you for your answer.  The comparison is fair to me.
> > > > I had to read again the paper to check FEAT, but now I recall the differences with your method.
> > > >
> > > > Going back to table 1. The main difference in performance is noticeable for tieredImageNet 5-shots, with Conv4-512 and ResNet-12. From the ablation, in figure-2a the main contribution to the improvement wrt the baseline protonet seems to be given by CEAM, while CECR gives you the extra point to beat FEAT. It would be more interesting to have the same ablation of figure 2a for tieredImageNet 5-shots with resnet12 and conv4-512 since there is much more margin to see the actual contribution of each component.
> > > >
> > > > I'm not here to stress the importance of top performance, I just think that this analysis will put your contribution in a much better light. Sorry for bringing this point so late in the reviewing process.

---

> > > > > ### Author Response · Authors · 2020-11-25
> > > > > **Response to Further Question on Ablation Study of AnonReviewer1**
> > > > >
> > > > > **1. It would be more interesting to have the same ablation of figure 2a for tieredImageNet 5-shots with resnet12 and conv4-512 since there is much more margin to see the actual contribution of each component.** \
> > > > > A: We are sorry for being unable to present the required results since *tiered*ImageNet is too large and there is no enough time for us to run the experiments. Instead, we conduct the suggested ablation study on the fine-grained CUB dataset. The obtained ablation results are given in the table below (which have also been added in Table 3 of Appendix A.4).
> > > > >
> > > > > | Method | Backbone | 5-way 1-shot | 5-way 5-shot |
> > > > > | - | :-: | :-: | :-: |
> > > > > | ProtoNet | Conv4-64 | 64.42$\pm$0.48 | 81.82$\pm$0.35 |
> > > > > | ProtoNet+CEAM | Conv4-64 | 68.92$\pm$0.50 | 84.54$\pm$0.32 |
> > > > > | MELR | Conv4-64 | 70.26$\pm$0.50 | 85.01$\pm$0.32 |
> > > > >
> > > > > We can see that: (1) Adding our CEAM to the baseline ProtoNet improves the performance by a large margin (2.7% - 4.5%) and our MELR further achieves noticeable improvements (0.5% - 1.3%) by adding CECR on top of ProtoNet+CEAM. This indicates that both CEAM and CECR are crucial for fine-grained FSL. (2) CEAM does make greater contribution to the final FSL results than CECR. This is expected since CECR is essentially a consistency constraint which does not have any learnable parameters.
> > > > >
> > > > > Moreover, we can also make similar observations with the ablation results under transductive FSL in Table 7 of Appendix A.10, where CECR achieves noticeable improvements (0.6% - 1.4%) on top of CEAM but again contributes relatively less to the final FSL performance than CEAM does (2.5% - 4.8% improvements over the baseline Semi-ProtoNet). Please also refer to Appendix A.10 for more details.

---

### Official Review · AnonReviewer4 · 2020-11-01
**The contribution of the paper is clear and novel. My initial rating is accept.**

**Rating:** 7
**Confidence:** 5

**Review:**

Summary

One problem of few-shot episodic learning is a poor sampling resulting in negative impacts on the learned model.
The paper proposes a new episodic training by exploiting inter-episode relationships to deal with poor sampling problem and improve the learned model by enforcing consistency regularization. Cross Episode Attention Module (CEAM) is proposed to alleviate the effect of poorly-sampled shots and Cross-Episode Consistency Regularization (CECR) is proposed to enforce robustness of the classifiers.

Strength

- The paper proposes a novel idea of how to improve few-shot learning by exploiting inter-episode relationships. Using multiple episodes and exploiting inter-episode is a new attempt.
- There have been attempts to improve few-shot training by batch construction, but the proposed method outperforms the previous approaches with a sizable margin.
- The extensive ablative studies provide comprehensive comparisons among possible design choices. (including supplementary materials)

Weakness

- I could not find a significant weakness of the paper.


Rating

I like the overall idea of using inter-episode relationships for few-shot training. The proposed approach shows strong performance and technically straightforward and easy to understand. Another strength of the method is that no additional hyper-parameter is used to tune the performance. The paper is clear and extensive experiments support the effectiveness of the paper including supplementary materials.

---

> ### Author Response · Authors · 2020-11-19
> **Response to AnonReviewer4**
>
> We’d like to greatly thank the reviewer for the positive comments!

---

### Decision · Program_Chairs · 2021-01-07
**Final Decision**

**Decision:**

Accept (Poster)

**Comment:**

This paper explores the effect of poorly sampled episodes in few-shot learning, and its effect on trained models. The improvements from the additional attention module (CEAM) and regularizer (CECR) are strong, and the ablations are thorough. The reviewers are not fully convinced that poor sampling is indeed the main issue. That is, it could be that CEAM and CECR improve performance for other reasons, but the hypothesis is sensible, and the reviewers believe a more thorough investigation is beyond the scope of this work.

During discussions, one note that came up is whether CEAM works because of cross-episode attention, or if the idea of an instance-level FEAT is itself a good one. One ablation to sort this out would be to apply FEAT and an instance-level FEAT on episodes that are twice as large as those seen by CEAM so that the effective episode size is the same. This would help answer: is it the reduced noise due to effectively larger episodes, a stronger attention mechanism using instance-level information, or is the idea of crossover episodes indeed the important factor? The reviewers agree that this baseline, or an analogous baseline, should be included in the final version.